# Bayesian Clustering of Neural Spiking Activity Using a Mixture of Dynamic Poisson Factor Analyzers

**Ganchao Wei**
Department of Statistics
University of Connecticut
ganchao.wei@uconn.edu

**Ian. H. Stevenson**
Department of Psychological Sciences
University of Connecticut
ian.stevenson@uconn.edu

**Xiaojing Wang**
Department of Statistics
University of Connecticut
xiaojing.wang@uconn.edu

## Abstract

Modern neural recording techniques allow neuroscientists to observe the spiking activity of many neurons simultaneously. Although previous work has illustrated how activity within and between known populations of neurons can be summarized by low-dimensional latent vectors, in many cases what determines a unique population may be unclear. Neurons differ in their anatomical location, but also, in their cell types and response properties. Moreover, multiple distinct populations may not be well described by a single low-dimensional, linear representation. To tackle these challenges, we develop a clustering method based on a mixture of dynamic Poisson factor analyzers (mixDPFA) model, with the number of clusters treated as an unknown parameter. To do the analysis of DPFA model, we propose a novel Markov chain Monte Carlo (MCMC) algorithm to efficiently sample its posterior distribution. Validating our proposed MCMC algorithm with simulations, we find that it can accurately recover the true clustering and latent states and is insensitive to the initial cluster assignments. We then apply the proposed mixDPFA model to multi-region experimental recordings, where we find that the proposed method can identify novel, reliable clusters of neurons based on their activity, and may, thus, be a useful tool for neural data analysis.

## 1 Introduction

With modern high-density probes (Jun et al., 2017), neuroscientists can observe the spiking activity of many neurons from many different anatomical regions simultaneously. With these expanding capabilities, new methods to analyze neural data at the population-level and at the level of multiple populations become necessary. Several recent models have been developed to extract shared latent structures from simultaneous neural recordings, assuming that neural activity can be described through low-dimensional latent states. Many existing approaches are extensions of two basic models: the linear dynamical system (LDS) model (Macke et al., 2011) and a Gaussian process factor analysis (GPFA) model (Yu et al., 2009). The LDS model is built on the state-space model and assumes latent factors evolve with linear dynamics. On the other hand, GPFA models the latent vectors by non-parametric Gaussian processes. However, in both cases, the observation model is generalized linear. Several variants of these models have been implemented to analyze multiple neural populations and their interactions (Semedo et al., 2019; Glaser et al., 2020). However, in many cases, the total number of distinct populations and which neurons belong to a population is unclear.

36th Conference on Neural Information Processing Systems (NeurIPS 2022).

Neurons in different anatomical locations may interact with each other or receive common input from unobserved brain areas, sharing the same latent structure. On the other hand, neurons of different cell-types within the same brain area may be better described by distinct latent structures. From a functional point of view, neither the anatomical location nor cell type (Fig. 1A) indicates which neurons should be grouped into the same populations. The incorrect population assignments can lead to biased and inconsistent inference on the latent structure (Ventura, 2009). If we instead ignore multi-population structure and treat all neurons as a single population, then using linear model based methods may not describe their activity well, especially when the input is non-homogeneous. Besides, nonlinear models such as deep learning (Pandarinath et al., 2018; Whiteway et al., 2019) and Gaussian processes (Wu et al., 2017) have been developed, but these models do not explicitly distinguish among distinct populations of neurons.

Motivated by the mixture of (Gaussian) factor analyzers (MFA,Arminger et al. 1999; Ghahramani and Hinton 1996; Fokoué and Titterington 2003), which describes globally nonlinear data by combining a number of local factor analyzers, here we group neurons based on the latent factors (Fig. 1B). A similar idea was previously implemented using a mixture of Poisson linear dynamical system (PLDS) model (mixPLDS, Buesing et al. 2014). The mixPLDS model infers the subpopulations and latent factors using deterministic variational inference Wainwright and Jordan (2008); Jordan et al. (1999); Emtiyaz Khan et al. (2013) and the model parameters are estimated by Expectation Maximization (EM). Unlike MFA, the mixPLDS can capture temporal dependencies of neural activity as well as interactions between clusters over time. However, there are several limitations for mixPLDS: 1) it requires we predetermine the number of clusters, and 2) the clustering results are often sensitive to the initial cluster assignment.

Here we cluster the neurons by a mixture of dynamic Poisson factor analyzers (mixDPFA). The DPFA model takes the advantages of both Poisson factor analysis (FA) and PLDS and includes both a population baseline and baselines for individual neurons. The number of clusters is treated as an unknown parameter in the mixDPFA, and the posteriors are sampled using Markov Chain Monte Carlo (MCMC). To sample high dimensional latent factors, we approximate the full conditional distribution of the latent state by a Gaussian, which is similar to results by sampling from exact full conditional distribution. To improve mixing in the cluster assignments, we marginalize the loading out for clustering by Poisson-Gamma conjugacy. We also discuss the constraints necessity for successful sampling of the proposed models. After validating the proposed model with simulated data, we apply it to analyze multi-region experimental recordings from behaving mice: the Visual Coding - Neuropixels Dataset from the Allen Institute for Brain Science. Overall, the proposed method provides a way to efficiently cluster neurons into populations based on their activity.

## 2 Methods

Here we introduce a mixture of dynamic Poisson factor analyzers (mixDPFA) to cluster neurons based on multi-population latent structure. The number of mixture components is treated as an unknown parameter and the posteriors are sampled by MCMC. In this section, we first provide the single population DPFA for a given cluster. Then, we introduce a prior on the number of clusters and describe how we use the mixture of finite mixture model (MFM) to efficiently sample the posterior of the mixDPFA.

### 2.1 Dynamic Poisson Factor Analyzer

Denote the observed spike count of neuron $i \in \{1, \ldots, N\}$ at time bin $t \in \{1, \ldots, T\}$ as $y_{it}$ (a non-negative integer), and let $\boldsymbol{y}_i = (y_{i1}, \ldots, y_{iT})'$. Further, let $z_i$ be the cluster indicator of neuron $i$. Motivated by the nature of neural activity and the former PLDS model (Macke et al., 2011), we propose a new Poisson FA model by adding individual baselines $\delta_i$. The proposed model is a combination of PLDS and Poisson FA, which includes both population baseline and individual baseline. Assume neuron $i$ belongs to the $j$-th cluster (i.e., $z_i = j$), and its spiking activity is independently Poisson distributed, conditional on the low-dimensional latent state $\boldsymbol{x}_t^{(j)} \in \mathbb{R}^{p_j}$ and population baseline $\mu_t^{(j)}$ as follows:

$$y_{it} \sim Poi(\lambda_{it}),$$
$$\log \lambda_{it} = \delta_i + \mu_t^{(j)} + \boldsymbol{c}_i' \boldsymbol{x}_t^{(j)},$$

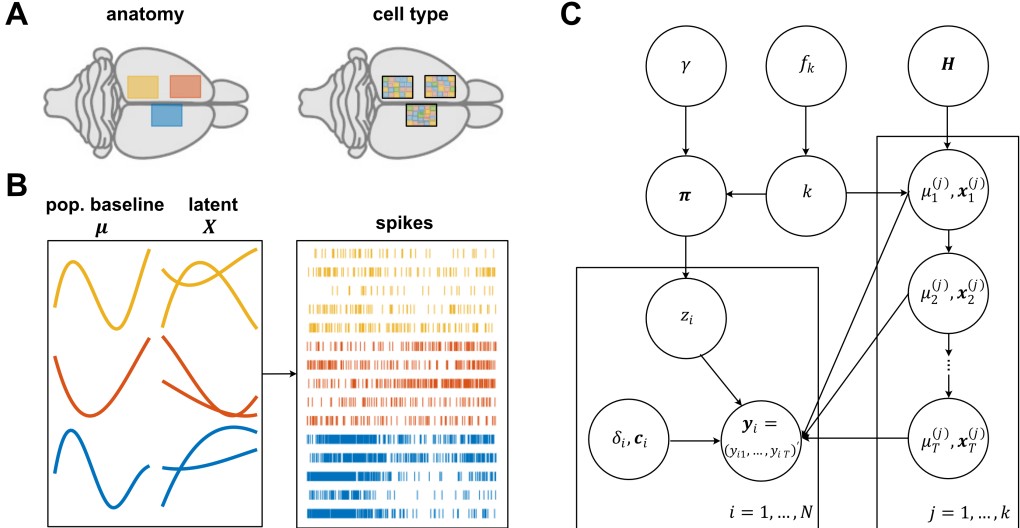

Figure 1: **Model overview. A.** There are multiple potential ways to define neural populations. For instance, populations could be defined by anatomical regions (left) or by cell types (right). Since the same latent structure could be shared across anatomical sites and cell types, a useful alternative may be define populations based on neural activity directly. **B.** The main goal for the proposed method is to cluster neurons according to their activity and extract functional grouping structure, based on spike train observations. The activity of each neuron is determined by a low dimensional latent state, specific to that neuron's cluster assignment (e.g. yellow, red, blue). **C.** Graphical representation of the mixture of finite mixtures (MFM) of dynamic Poisson factor analyzers (DPFA) generative model. Here the cluster number is treated as a random variable. The population baseline ($\mu_t^{(j)}$) and the latent factor ($\boldsymbol{x}_t^{(j)}$) for each cluster is generated by linear dynamics, with a Gaussian noise.

with $\boldsymbol{c}_i \sim \mathcal{N}_{p_j}(\mathbf{0}, \boldsymbol{I}_{p_j})$. The neuron-specific baseline $\delta_i$ is a constant across time for the $i$th neuron and unrelated to the cluster assignment. For simplicity, we assume the dimension of latent factors (states) is the same for all clusters, s.t. $p_j = p$, however, our method easily extends to the situation when $p_j$s differs across clusters (see Discussion). Further, we assume the population baseline $\mu_t^{(j)}$ and the latent state $\boldsymbol{x}_t^{(j)}$ evolve linearly over time with Gaussian noise as following

$$\mu_{t+1}^{(j)} = g^{(j)} + h^{(j)}\mu_t^{(j)} + \epsilon_t^{(j)},$$
$$\boldsymbol{x}_{t+1}^{(j)} = \boldsymbol{b}^{(j)} + \boldsymbol{A}^{(j)}\boldsymbol{x}_t^{(j)} + \boldsymbol{\eta}_t^{(j)},$$

where $\epsilon_t^{(j)} \sim \mathcal{N}(0, \sigma^{2(j)})$ and $\boldsymbol{\eta}_t^{(j)} \sim \mathcal{N}_p(\mathbf{0}, \boldsymbol{Q}^{(j)})$.

If we denote $\boldsymbol{\lambda}_i = (\lambda_{i1}, \dots, \lambda_{iT})'$, $\boldsymbol{\mu}^{(j)} = (\mu_1^{(j)}, \dots, \mu_T^{(j)})'$ and $\boldsymbol{X}^{(j)} = (\boldsymbol{x}_1^{(j)}, \dots, \boldsymbol{x}_T^{(j)})'$, the proposed model can be rewritten as

$$\begin{aligned} \boldsymbol{y}_i &\sim Poi(\boldsymbol{\lambda}_i), \\ \log \boldsymbol{\lambda}_i &= \delta_i \mathbf{1}_T + \boldsymbol{\mu}^{(j)} + \boldsymbol{X}^{(j)} \boldsymbol{c}_i. \end{aligned} \tag{1}$$

Generally, a factor model is consistent only when $T/N \to 0$ (Johnstone and Lu, 2009), but this is often not the case for most neural spike data. However, when we assume linear dynamics on $\boldsymbol{\mu}^{(j)}$ and $\boldsymbol{X}^{(j)}$, it resolves the consistency issue. As known in a FA model, when $p > 1$, the model is only identifiable up to orthogonal rotation on $\boldsymbol{X}^{(j)}$, with $\boldsymbol{c}_i \sim N(\mathbf{0}, \boldsymbol{I}_p)$. With including an individual baseline $\delta_i \mathbf{1}_T$ in our proposed DPFA model (1), it further makes the model invariant to translation of $\boldsymbol{\mu}^{(j)}$ and $\boldsymbol{X}^{(j)}$. That means if $\boldsymbol{\mu}^{(j)}$ and $\boldsymbol{X}^{(j)}$ is a set solution, then $\boldsymbol{\mu}^{(j)} + a\mathbf{1}_T$ and $\boldsymbol{X}^{(j)}\boldsymbol{U} + \mathbf{1}_T \otimes \boldsymbol{m}'$ also satisfy the model, for any $a$, $\boldsymbol{m}$ and orthogonal matrix $\boldsymbol{U}$. Thus, to make the model identifiable, we need to add several constraints. Although the clustering is invariant to orthogonal rotation, how we put constraints on translation will influence the cluster assignments. Here we assume $\boldsymbol{A}^{(j)}$ and

$Q^{(j)}$ are diagonal (Peña and Poncela, 2004; Lopes et al., 2008), and, to encourage clustering based on the trajectories of latent factors, we set $\sum_{t=1}^{T} \mu_t^{(j)} = 0$ and $\sum_{t=1}^{T} x_t^{(j)} = 0$. See Section 5 for more discussions about the choice of these constraints.

Given the parameters of the $j$-th cluster $\theta^{(j)} = \{\mu^{(j)}, X^{(j)}, h^{(j)}, g^{(j)}, \sigma^{2(j)}, A^{(j)}, b^{(j)}, Q^{(j)}\}$, the spike counts of neuron $i$ are generated by the dynamic Poisson factor analyzer (DPFA) model as $[y_i \mid z_i = j] \sim DPFA(\delta_i, c_i, \theta^{(z_i)})$. To faciliate the Bayesian computation, we have to impose priors $H$ on $\theta^{(j)}$, see more details of prior settings in Appendix .

## 2.2 Clustering by Mixture of Finite Mixtures Model

When the population labels $z_i$s are unknown, we cluster the neurons by a mixture of DPFA (mixDPFA). Since the number of neural populations is finite but unknown, we need to put priors on it. To make the Bayesian computation more efficient, we utilize the idea from the mixture of finite mixtures (MFM, Miller and Harrison 2018) model, by assigning the priors for the clusters in the following way:

$$
\begin{aligned}
k &\sim f_k, & f_k \text{ is a p.m.f. on} \{1, 2, \ldots\}, \\
\pi = (\pi_1, \ldots, \pi_k) &\sim Dir_k(\gamma, \ldots, \gamma) & \text{given } k, \\
z_1, \ldots, z_N &\overset{i.i.d.}{\sim} \pi & \text{given } \pi, \\
\theta^{(1)}, \ldots, \theta^{(k)} &\overset{i.i.d.}{\sim} H & \text{given } k, \\
y_i = (y_{i1}, \ldots, y_{iT})' &\sim \text{DPFA}(\delta_i, c_i, \theta^{(z_i)}) & \text{given } \delta_i, c_i, \theta^{(z_i)}, z_i, \forall i = 1, \ldots, N,
\end{aligned}
\tag{2}
$$

where p.m.f denotes the probability mass function. By using the MFM, we can integrate the field knowledge about the number of neural populations into our analysis. In the analysis of this paper, we assume $k$ follows a geometric distribution, i.e., $k \sim Geometric(\alpha)$ with its density defined as $f_k(k|\alpha) = (1 - \alpha)^{k-1}\alpha$ for $k = 1, 2, \ldots$, and let $\gamma = 1$. The complete generative model is summarized in a graphical form shown in Fig. 1C.

## 2.3 Inference

Here the posteriors of the proposed mixDPFA model are sampled by an MCMC algorithm (see Appendix A.1). In each iteration, we sample the model parameters assuming the known cluster indices at first, and then sample the cluster indices given the model parameters. When sampling the (labeled) model parameters, the latent state $X^{(j)}$ and population baseline $\mu^{(j)}$ have no closed-form full conditional distributions. Although we could sample the posterior by particle MCMC directly, convergence may be too slow for clustering. Here, we approximate the full conditional distribution for $X^{(j)}$ and $\mu^{(j)}$ by a Gaussian distribution (a Laplace approximation) and generate samples according to this approximation. This Laplace approximation is widely used for EM (Macke et al., 2011) and variational inference (Glaser et al., 2020) with PLDS models and their variants, and Gaussian approximation of the intractable full conditional distribution of latent effects has also been used in Bayesian mixed effects binomial regression, where Berman et al. 2022 found that the approximation provided reasonable estimation accuracy with substantial computational speedups. We, thus, use a global Laplace approximation that can be efficiently computed in $\mathcal{O}(T)$ (Paninski et al., 2010). To help convergence, sampling on (labeled) model parameters is repeated several times before updating the cluster indices. Approximating the intractable full conditional with a Laplace approximation also makes computation of the proposed mixDPFA more efficient. However, to assess the accuracy of the approximation we also compare our approach to directly sampling from the exact posterior of the model. We develop a Pólya-Gamma (PG) data augmentation approach (Windle et al., 2013; Linderman et al., 2017, 2016; Polson et al., 2013) with an additional Metropolis-Hastings (MH) step (Metropolis et al., 1953; Hastings, 1970) to sample exactly from the full conditional of $X^{(j)}$ and $\mu^{(j)}$. We find that the proposed method using a Laplace approximation is faster but performs similarly as sampling from the exact joint posterior (Fig. 2, Fig. 1, Appendix A.2, and Table 1)).

Once we update the latent state $X^{(j)}$ and population baseline $\mu^{(j)}$, the cluster index is then sampled by the analogy of partition-based algorithm in Dirichlet process mixtures (DPM, Neal 2000). See details in Miller and Harrison 2018 and the Appendix (A.1). When doing the clustering, we need to evaluate the likelihood for neurons under each cluster. Although we can sample $c_i$ directly and evaluate the full likelihood as in MCMC for Gaussian MFA (data-augmentation/ imputation-posterior

algorithm, Fokoué and Titterington 2003), the chain has poor mixing and stops after a few iterations, because of the high dimensionality. The heavy dependency on the starting point when fitting the mixture of PLDS (mixPLDS, Buesing et al. 2014) model may suggest a similar problem. To resolve this, we evaluate the marginal likelihood by integrating out the neuron-specific $c_i$, i.e., the marginal likelihood of neuron $i$ in cluster $j$ is computed by

$$M_{\boldsymbol{\theta}^{(j)}}(\boldsymbol{y}_i) = P(\boldsymbol{y}_i|\boldsymbol{\theta}^{(j)}, \delta_i) = \int P(\boldsymbol{y}_i|\boldsymbol{\theta}^{(j)}, \delta_i, \boldsymbol{c}_i) P(\boldsymbol{c}_i) \, d\boldsymbol{c}_i. \tag{3}$$

However, this marginal likelihood has no closed form. Though we may evaluate it by a Laplace approximation, but iterating over all potential clusters for each neuron is computationally intensive. To make faster clustering, we approximate the marginal likelihood by utilizing a Poisson-Gamma conjugacy. This approach has been previously utilized to approximate posteriors (El-Sayyad, 1973) and predictive distributions (Chan and Vasconcelos, 2009). In our situation, since $\boldsymbol{c}_i \sim \mathcal{N}(\boldsymbol{0}, \boldsymbol{I}_p)$, we have $\lambda_{it} = \exp(\delta_i + \mu_t^{(j)} + \boldsymbol{c}_i' \boldsymbol{x}_t^{(j)}) \sim lognormal(\delta_i + \mu_t^{(j)}, \boldsymbol{x}_t'^{(j)} \boldsymbol{x}_t^{(j)})$, and then we can approximate this lognormal distribution by a gamma distribution, i.e., assume $\lambda_{it}$ follows $Gamma(a_{it}, b_{it})$ with $a_{it} = (\boldsymbol{x}_t'^{(j)} \boldsymbol{x}_t^{(j)})^{-1}$ and $b_{it} = \boldsymbol{x}_t'^{(j)} \boldsymbol{x}_t^{(j)} \cdot e^{\delta_i + \mu_t^{(j)}}$. Then, by the conjugate property with Poisson and Gamma random variables, we have

$$P(y_{it}|\boldsymbol{\theta}^{(j)}, \delta_i) = \int P(y_{it}|\lambda_{it}) P(\lambda_{it}) \, d\lambda_{it} \approx NB(y_{it}|\nu_{it}, p_{it}),$$

with $\nu_{it} = a_{it}$ and $p_{it} = 1/(1 + b_{it})$. Further, noticing that we have the conditional independence assumption for $P(\boldsymbol{y}_i|\boldsymbol{\theta}^{(j)}, \delta_i)$, that is $P(\boldsymbol{y}_i|\boldsymbol{\theta}^{(j)}, \delta_i) = \prod_{t=1}^T P(y_{it}|\boldsymbol{\theta}^{(j)}, \delta_i)$, we then have a closed-form for Equation (3).

Another possible idea is to approximate the log-likelihood by second-order polynomials, with coefficients determined by Chebyshev polynomial approximation (Keeley et al., 2020). However, we find that this approximation doesn't work well in practice when spike counts have a wide range. The model is implemented in MATLAB and the code is available at `https://github.com/weigcdsb/MFM_DPFA_clean`.

## 3 Simulations

To validate and illustrate the proposed clustering method, we simulate neural data directly from the generative model (1) . The labels for each neuron are assumed known and fixed at first to check convergence and model identifiability. We then infer the labels to evaluate clustering performance. All experiments in this paper were performed using a 3.40 GHz processor with 16 GB of RAM.

### 3.1 Labeled data

We first simulate 10 clusters with 5 neurons in each, with recording length $T = 1000$ and $p = 2$ dimensional latent factors for each cluster. Individual baselines are generated by $\delta_i \sim N(0, 0.5^2)$, and the loading for the latent states are generated by $\boldsymbol{c}_i \sim N(\boldsymbol{0}, \boldsymbol{I}_2)$. The population baseline $\boldsymbol{\mu}^{(j)}$ and latent vector $\boldsymbol{X}^{(j)}$ are generated by the spline interpolation on 10 to 30 evenly spaced knots. The simulations are conducted 50 times with different seeds. Here we show results for one simulation, and the performance for the rest is similar.

Since the labels are known, each whole simulation is equivalent to 10 independent simulations, with 5 neurons in each. Running MCMC for 10,000 iterations, we find that the log-likelihood per spike converges rapidly for individual clusters and overall (Fig.2A). Trace plots (Fig.2B) of the (Frobenius) norms for linear dynamics samples $(h^{(j)}, g^{(j)}, \sigma^{2(j)}, \boldsymbol{A}^{(j)}, \boldsymbol{b}^{(j)}, \boldsymbol{Q}^{(j)})$ show rapid mixing and convergence for each DPFA. The fitted mean firing rate (mean response) (Fig. 2C), $\boldsymbol{\mu}^{(j)}$ and $\boldsymbol{X}^{(j)}$ (Fig. 2D) match the ground truth well. The convergence is fast, especially in terms of mean response and population baseline $\boldsymbol{\mu}^{(j)}$. Together, these results demonstrate the identifiability of the DPFA with appropriate constraints.

We then compare the 10-cluster model with the simplified model ignoring the clustering structure (1-cluster model). The $p$ for 1-cluster model is 14, chosen by 5-fold speckled cross-validation described in (Williams et al., 2020). To evaluate the fitting performance of these two models, we

hold out 1/2 of the data and compare the held-out log-likelihood per spike. The distribution training log-likelihood is evaluated by averaging over the samples in short chains (e.g. iteration 50 to 100), which is justified by the observed fast convergence. The same procedure is replicated for 50 times. In this case, ignoring the clustering structure leads to a worse performance (Fig 2E), and the single population analysis cannot describe the data as accurately, since the input is non-homogeneous and the data is global nonlinear.

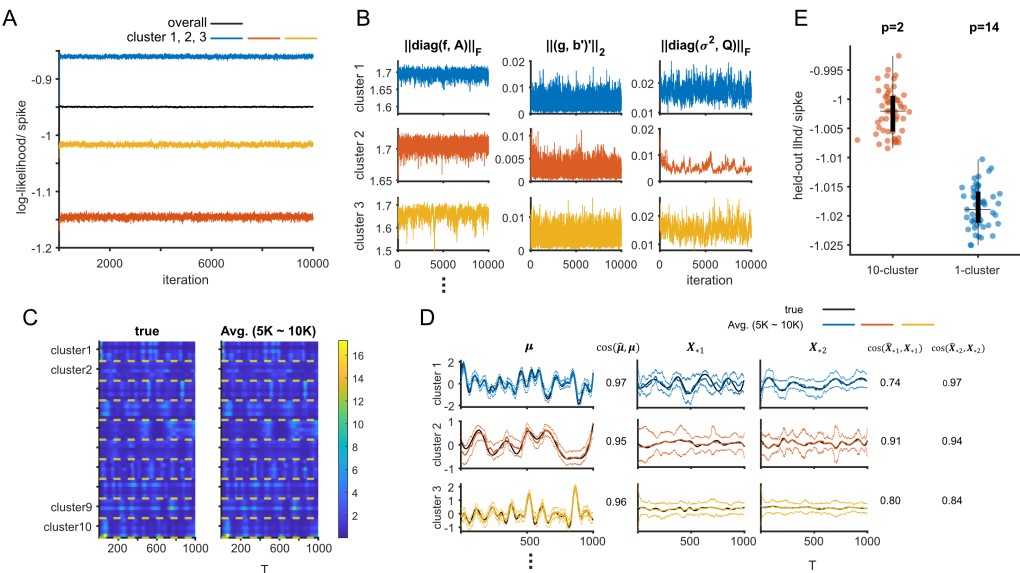

Figure 2: **Bayesian inference with labeled data** Here we simulate 10 clusters and assess convergence and mixing of for the DPFA. **A.** Traceplot of the log-likelihood per spike for all neurons and the first 3 clusters. **B.** The traceplot of (Frobenius) norms of linear dynamics of $\boldsymbol{\mu}^{(j)}$ and $\boldsymbol{X}^{(j)}$ for each cluster (showing the first 3). **C.** The true and the fitted mean firing rate, showing the averages over samples from iteration 5000 to 10000. **D.** The true (black) and the fitted (colored) population baseline and latent factor. The $\boldsymbol{X}_{*l}$ denotes the $l$-th latent factor (i.e. the $l$-th column of $\boldsymbol{X}$). The dashed lines show the 95% highest posterior density (HPD) interval. The cosine (the "overlap") between true values and posterior means shown besides. **E.** Comparison of the held-out likelihood per spike when fitting to 1/2 of the data: 1) 10-cluster model where each cluster has $p = 2$ (true value), and 2) 1-cluster model where a single DPFA describes all neurons, with $p = 14$ selected by 5-fold speckled cross-validation. Dots denote results from individual short (iterations 50-100), independent chains.

## 3.2 Clustering

Using the same simulation, we now infer the cluster labels. The latent factor dimension was first optimized with $p = 2$ selected by 5-fold speckled CV on small chains (100 iterations). We then compare three chains fitting with all data: two unique chains initialized using a single cluster and one chain initialized with $N = 50$ clusters (i.e. all clusters are singletons). Traceplots (Fig. 3A) of training log-likelihood and number of clusters show that all chains converge. When fit to the full data or only half, the number of clusters converges to 10, although the prior over the number of clusters is $K \sim Geometric(0.2)$. When the recording length is sufficiently long, the likelihood will dominate, and the number of clusters will not be much affected by the prior setting. The true mean firing rate for each neuron (Fig. 3B) can be well recovered, even with half data held out. To evaluate cluster membership, here we show a similarity matrix where the entry $(i, l)$ is the posterior probability that data points $i$ and $l$ belong to the same cluster. The clustering results for all 4 chains recover the true clusters, no matter what the the starting assignment is (Fig. 3C), which suggest the convergence of MCMC. The overall performance of the full model, where cluster membership is inferred alongside the latent states, is similar to the case when cluster labels are known and substantially higher than the 1-cluster model (Fig. 3D and E).

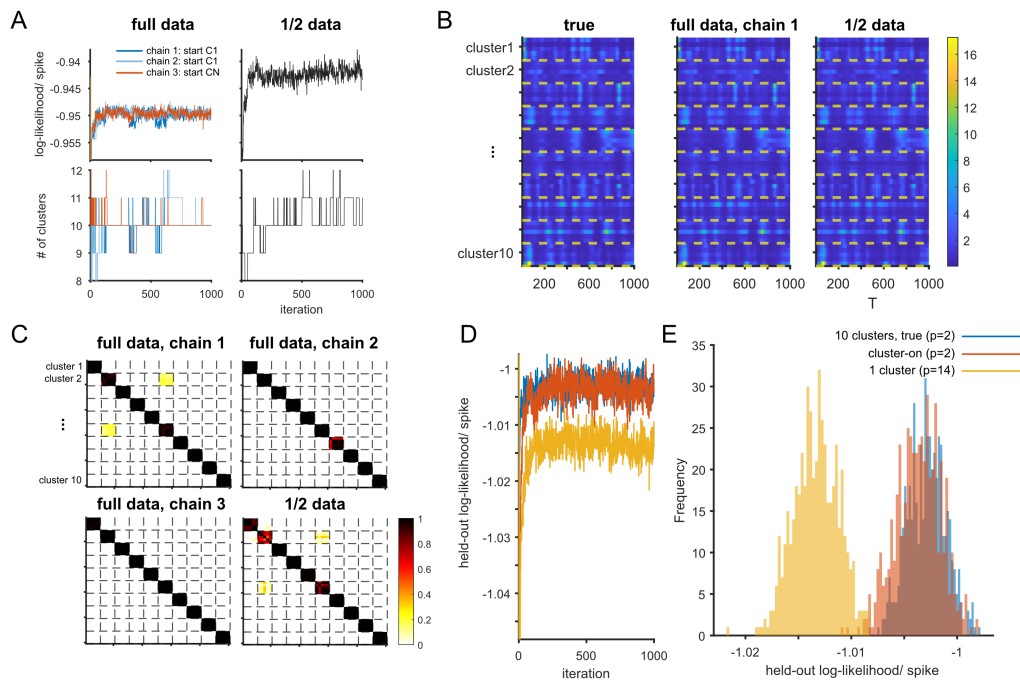

Figure 3: **Bayesian clustering** The same simulation setting as in Fig. 2, but inferring cluster labels from spike observations alone. **A.** The trace plots of training log-likelihood per spike and the number of clusters. The model is fitted by using all and half data as training with three chains shown for full data (initialized with a single cluster, C1 or $N = 50$ clusters, CN). **B.** The true and fitted mean firing rate, averaging from iteration 500 to 1000. **C.** The posterior similarity matrix for all chains (rows and columns ordered according to the ground truth). **D.** Traceplot of the held-out (1/2) log-likelihood for 1) the 10-cluster model with labels known, 2) the full model estimating labels and optimizing $p$, and 3) the 1-cluster, single population model with $p = 14$ chosen by CV. **E.** Held-out log-likelihood per spike for each model (samples for iteration 500 to 1000).

# 4 Multi-region neural spike recordings

We then apply the proposed clustering method to the Allen Institute Visual Coding Neuropixels dataset. The dataset contains spiking activity from hundreds of neurons from multiple brain regions of an awake mouse. See detailed data description in (Siegle et al., 2021). Here we investigate the clustering structure of neurons from four anatomical sites (83 neurons): 1) hippocampal CA1 (24 neurons), 2) dorsal part of the lateral geniculate complex (LGd, 36 neurons), 3) lateral posterior nucleus of the thalamus (LP, 12 neurons) and 4) primary visual cortex (VISp, 11 neurons). And we analyze responses to 20s epochs during three visual stimuli: drifting gratings, spontaneous activity, and natural movies. Only neurons with rates > 1Hz within the selected epochs are included (72% of 115 neurons) and we analyze data with 40ms bins. We use a $Geometric(0.33)$ prior over the number of clusters, such that $p(k \leq 4) = 0.8$.

In responses to drifting gratings, the 5-fold speckled CV log-likelihood is optimized with $p = 2$, and, as in the simulations, the log-likelihood and number of clusters show rapid convergence and mixing (Fig 4A and B). Low firing rates and short recording lengths tend to cause confusions in clustering, reflecting uncertainty in cluster membership for neurons with little information. Here the average number of clusters is 16. To summarize the clustering results stored as posterior samples in MCMC, we give the single estimate for cluster indices $\hat{z}_i$ by maximizing the posterior expected adjusted Rand index (maxPEAR, Fritsch and Ickstadt 2009). The maxPEAR-sorted neural activity and posterior similarity matrix are shown in Fig. 4C and D. Results sorted by Maximum a posteriori (MAP) estimate are similar and are shown in the Appendix (Fig. 5). To examine the relationship between the clustering results and anatomy, we additionally sort the neurons according anatomical labels (upper left panel in Fig. 4E). Although many identified clusters are neurons from the same anatomical area,

clusters also include neurons from different regions and neurons within a region are often clustered into separate populations ($\hat{P}(\text{neuron } i, l \text{ in the same region}|z_i = z_l, \{\boldsymbol{y}_i\}_{i=1}^{N}) = 0.57$). Together, these results suggest that a simple assignment of populations based on anatomy many not accurately represent the latent structure.

We then evaluate the clustering patterns for different visual stimuli. We run 2 independent chains for each epoch (results from the second chain in Fig. 2D). The similarity matrices show that the pattern is consistent for the same epoch, but will change along the time even under the same experimental settings (D1 vs. D2 and S1 vs. S2). The changes in the clustering patterns may suggest long-term drift for neuron interactions. To quantify the observations, we evaluate the adjusted Rand index (ARI) of maxPEAR estimates (bottom right panel in Fig. 4E). Between-epoch comparisons tend to have lower similarity (average ARI from comparing 2 chains for each epoch) than within-epoch comparisons (different chains) for both maxPEAR and MAP (Fig. 2C).

The MAP number of clusters is largest (18) for the natural movie, suggesting this epoch has the most severe global non-linearity issue. Here we compare three models: 1) clustering model with $p = 2$, 2) single cluster model, with $p = 8$ selected by 5-fold speckled CV and 3) anatomical cluster model. The anatomical cluster model fits a single DPFA for each region, using the anatomical labels to define the clusters explicitly. Using cross-validation, we find that the optimized dimension for each region is $p = (1, 11, 5, 3)$, respectively. We find that the single cluster model tends to underfit the data, while the anatomical cluster model tends to overfit the data (Fig. 4F).

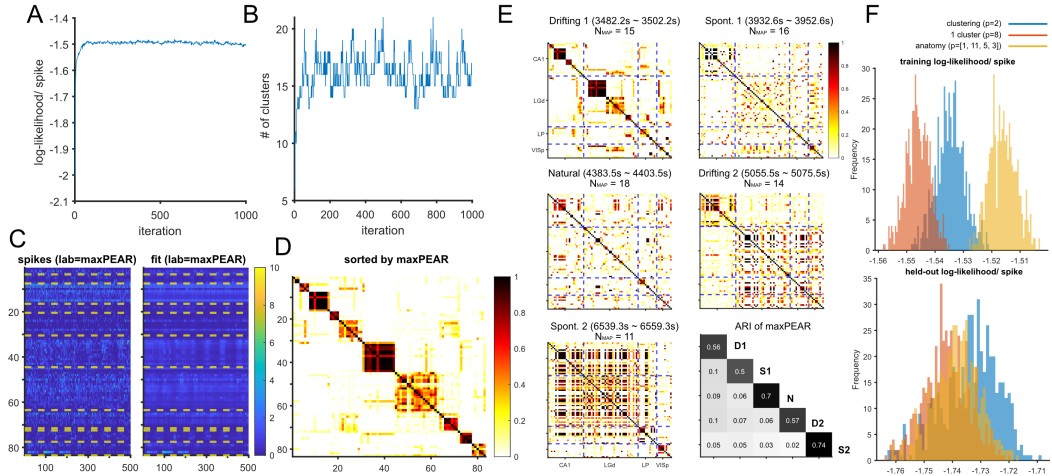

Figure 4: **Application in Neuropixels data. A. and B.** The trace plots of log-likelihood per spike and number of clusters for drifting grating responses. All results are averages from iteration 500 to 1000. **C.** The observed spikes and fitted mean firing rate, sorted by the maxPEAR label. **D.** The posterior similarity matrix, sorted by the maxPEAR label. **E.** The posterior similarity matrices for 4 adjacent epochs and 1 further epoch with different visual stimuli, sorted sorted according to maxPEAR estimate and anatomical label in the first drifting grating epoch. The last panel shows the adjusted Rand index of the maxPEAR estimates. The diagonal is the ARI between two chains for the same data, while off-diagonal values show the mean ARI of maxPEAR for the four comparisons between two chains from two different epochs. **F.** For the natural epoch, we hold out 1/2 data as the training, and show the histograms of training and held-out log-likelihood per spike, from iteration 500 to 1000 for three models: 1) clustering model, 2) single cluster model, and 3) anatomical cluster model.

## 5   Discussion

Here we introduce a Bayesian approach to cluster neural spike trains by MCMC. Previous approaches to multi-population latent variable modeling have used anatomical information to label distinct groups of neurons, but this choice is somewhat arbitrary. Brain region and cell-type, for instance, can give

contradictory population labels. The proposed method groups neurons by common latent factors, which may be useful for identifying "functional populations" of neurons. Here we use a mixDPFA model and infer the number of clusters by MFM with a partition-based algorithm similar to DPM. MFM may be more conceptually appropriate than DPM, since the number of neural "populations" is unknown but finite. Additionally, MFM produces more concentrated, evenly dispersed clusters (see Miller and Harrison 2018 for detailed discussion). The mixture modeling approach may also be appropriate in cases where neurons share non-homogeneous inputs, since it can approximate global nonlinearity with a mixture of locally linear models. Here we find that the mixture model outperforms globally linear (1-cluster) models in simulations and with experimental data.

Although the proposed method can describe data and cluster neural spiking activity successfully, there are some potential improvements. Firstly, as mentioned above, the unconstrained model does not have unique solutions. To ensure model identifiable, we put diagonal constraints on $A^{(j)}$ and $Q^{(j)}$ and constrain $\mu^{(j)}$ and $X^{(j)}$ to have mean zero. The assumption that $A^{(j)}$ and $Q^{(j)}$ are diagonal does not allow interaction between latent factors. However, these interactions could be allowed by instead constraining $X'^{(j)}X^{(j)}$ to be diagonal (Krzanowski and Marriott, 1994b,a; Fokoué and Titterington, 2003). Such a constraint could allow unique solutions for the (P)LDS and GPFA. A second potential improvement would be to automatically infer the dimension of the latent factors (states). In this paper, we assume $p_j$ is the same for all clusters, for convenience. $p$ is a pre-selected value or can be selected by cross-validation (CV). This may limit the accuracy of the model, since populations of neurons in experimental data are likely to have different latent dimensionalities. In future work, it would also be possible to treat $p_j$ as a parameter and sample the posterior by a reversible-jump (RJ)MCMC (Lopes and West, 2004), birth-death (BD)MCMC (Stephens, 2000; Fokoué and Titterington, 2003), or adaptive Gibbs sampling with shrinkage prior on $X^{(j)}$ (Bhattacharya and Dunson, 2011). Although RJMCMC and BDMCMC can be easily implemented, they perform poorly for high dimensional data and may be sensitive to priors. Adaptive Gibbs sampling with shrinkage, on the other hand, has been implemented with the infinite mixture of infinite factor analyzers (IMIFA, Murphy et al. 2020). The same idea may be useful here with an additional prior on linear dynamics ($A^{(j)}$, $b^{(j)}$ and $Q^{(j)}$) to encourage shrinkage in $X^{(j)}$. Finally, a deterministic approximation of MCMC, such as variational inference may be more computationally efficient. Standard methods for fitting the PLDS could be used directly in the VI updates, and if we further use a stick-breaking representation for the MFM model, it would be straightforward to use VI for clustering as well, similar to (Blei and Jordan, 2006).

As the number of neurons and brain regions that neuroscientists are able to record simultaneously continues to grow, understanding the latent structure of multiple populations will be a major statistical challenge. The Bayesian approach to clustering neural spike trains introduced here converges fast and is insensitive to the initial cluster assignments, and may, thus, be a useful tool for identifying "functional populations" of neurons.

## Acknowledgments

This material is based upon work supported by the National Science Foundation under Grant No. 1931249.

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
