# Appendix for Bayesian Clustering of Neural Spiking Activity Using a mixDPFA

## A  Appendix

### A.1  MCMC updates

The posteriors are sampled by a Gibbs sampler. In each iteration, the sampling scheme has two main stages: 1) to sample the model parameters assuming known labels and 2) to sample the cluster indices given model parameters. Before moving into the second stage, the first stage is repeated several times. The sampling in the first stage is conducted without considering constraints for $\mu_t^{(j)}$ and $\boldsymbol{x}_t^{(j)}$ at first, and then we project the samples onto the constraint space for $\sum_{t=1}^{T} \mu_t^{(j)} = 0$ and $\sum_{t=1}^{T} \boldsymbol{x}_t^{(j)} = \boldsymbol{0}$ as used in (Sen et al., 2018).

**Update $\boldsymbol{x}_t^{(j)}$ and $\mu_t^{(j)}$**    The priors for initial population baseline and latent factor are:

$$\mu_1^{(j)} \sim \mathcal{N}(0, 1),$$
$$\boldsymbol{x}_1^{(j)} \sim \mathcal{N}(\boldsymbol{0}, \boldsymbol{I}_p).$$

Assume there are $n_j = \#\{i : z_i = j\}$ neurons belong to the $j$-th cluster, and denote the spike counts and firing rates of these neurons at time $t$ as $\widetilde{\boldsymbol{y}}_t^{(j)} = vec(\{y_{it} | z_i = j\}) \in \mathbb{Z}_{\geq 0}^{n_j}$ and $\widetilde{\boldsymbol{\lambda}}_t^{(j)} = vec(\{\lambda_{it} | z_i = j\}) \in \mathbb{R}^{n_j}$, where $\mathbb{Z}_{\geq 0}^{n_j}$ denotes a $n_j$-dimensional vector with each element being non-negative integers. The corresponding loading and baseline for these $n_j$ neurons is $\boldsymbol{C}^{(j)} \in \mathbb{R}^{n_j \times p}$ and $\boldsymbol{\Delta}_j = vec(\{\delta_i | z_i = j\}) \in \mathbb{R}^{n_j}$, such that $\log \widetilde{\boldsymbol{\lambda}}_t^{(j)} = \boldsymbol{\Delta}_j + \mu_t^{(j)} \boldsymbol{1}_{n_j} + \boldsymbol{C}^{(j)} \boldsymbol{x}_t^{(j)} = \boldsymbol{\Delta}_j + \left(\boldsymbol{1}_{n_j}, \boldsymbol{C}^{(j)}\right) \left(\mu_t^{(j)}, \boldsymbol{x}_t'^{(j)}\right)'$. Denote $\widetilde{\boldsymbol{C}}^{(j)} = \left(\boldsymbol{1}_{n_j}, \boldsymbol{C}^{(j)}\right)$, $\widetilde{\boldsymbol{x}}_t^{(j)} = \left(\mu_t^{(j)}, \boldsymbol{x}_t'^{(j)}\right)'$, $\widetilde{\boldsymbol{x}}^{(j)} = \left(\widetilde{\boldsymbol{x}}_1'^{(j)}, \ldots, \widetilde{\boldsymbol{x}}_T'^{(j)}\right)'$, $\widetilde{\boldsymbol{A}}^{(j)} = diag(h^{(j)}, \boldsymbol{A}^{(j)})$, $\widetilde{\boldsymbol{b}}^{(j)} = (g^{(j)}, \boldsymbol{b}'^{(j)})'$ and $\widetilde{\boldsymbol{Q}}^{(j)} = diag(\sigma^{2(j)}, \boldsymbol{Q}^{(j)})$. The full conditional distribution $P(\widetilde{\boldsymbol{x}}^{(j)} | \ldots) = P(\widetilde{\boldsymbol{x}}^{(j)} | \{\widetilde{\boldsymbol{y}}_t^{(j)}\}_{t=1}^T, \boldsymbol{\Delta}_j, \widetilde{\boldsymbol{C}}^{(j)}, \widetilde{\boldsymbol{A}}^{(j)}, \widetilde{\boldsymbol{b}}^{(j)}, \widetilde{\boldsymbol{Q}}^{(j)})$ is approximated by a global Laplace approximation, i.e.,

$$P(\widetilde{\boldsymbol{x}}^{(j)} | \ldots) \approx \mathcal{N}_{(p+1)T}(\widetilde{\boldsymbol{x}}^{(j)} | \boldsymbol{\mu}_{\widetilde{\boldsymbol{x}}^{(j)}}, \boldsymbol{\Sigma}_{\widetilde{\boldsymbol{x}}^{(j)}}),$$
$$\boldsymbol{\mu}_{\widetilde{\boldsymbol{x}}^{(j)}} = \underset{\widetilde{\boldsymbol{x}}^{(j)}}{\arg\max} \, P(\widetilde{\boldsymbol{x}}^{(j)} | \ldots),$$
$$\boldsymbol{\Sigma}_{\widetilde{\boldsymbol{x}}^{(j)}} = -(\nabla\nabla \log P(\widetilde{\boldsymbol{x}}^{(j)} | \ldots)|_{\widetilde{\boldsymbol{x}}^{(j)} = \boldsymbol{\mu}_{\widetilde{\boldsymbol{x}}^{(j)}}})^{-1}.$$

Then, taking the logarithm of the full conditional distribution, i.e., $\ell = \ell(\widetilde{\boldsymbol{x}}^{(j)}) = \log P(\widetilde{\boldsymbol{x}}^{(j)} | \ldots)$, we have

$$\ell = \text{const} + \sum_{t=1}^{T} \left(\widetilde{\boldsymbol{y}}_t'^{(j)}(\boldsymbol{\Delta}_j + \widetilde{\boldsymbol{C}}^{(j)} \widetilde{\boldsymbol{x}}_t^{(j)}) - \boldsymbol{1}_{n_j}' \widetilde{\boldsymbol{\lambda}}_t\right) - \frac{1}{2}(\widetilde{\boldsymbol{x}}_1^{(j)} - \widetilde{\boldsymbol{x}}_0'^{(j)})[\widetilde{\boldsymbol{Q}}_0^{(j)}]^{-1}(\widetilde{\boldsymbol{x}}_1^{(j)} - \widetilde{\boldsymbol{x}}_0^{(j)})$$

$$- \sum_{t=2}^{T} \frac{1}{2}(\widetilde{\boldsymbol{x}}_t^{(j)} - \widetilde{\boldsymbol{A}}^{(j)} \widetilde{\boldsymbol{x}}_{t-1}^{(j)} - \widetilde{\boldsymbol{b}}'^{(j)})[\widetilde{\boldsymbol{Q}}^{(j)}]^{-1}(\widetilde{\boldsymbol{x}}_t^{(j)} - \widetilde{\boldsymbol{A}}^{(j)} \widetilde{\boldsymbol{x}}_{t-1}^{(j)} - \widetilde{\boldsymbol{b}}^{(j)}).$$

36th Conference on Neural Information Processing Systems (NeurIPS 2022).

Here, $\ell(\widetilde{\boldsymbol{x}}^{(j)})$ is concave and unimodal. By the Markovian assumption for latent state vectors, the Hessian matrix is tri-block diagonal. We can thus compute Newton updates to get $\boldsymbol{\mu}_{\widetilde{\boldsymbol{x}}^{(j)}}$ for the Laplace approximation in $\mathcal{O}(T)$ (Paninski et al., 2010), similar to the E-step for the PLDS (Macke et al., 2011).

The gradient $\nabla\ell$ and the Hessian $\nabla\nabla\ell$ are provided as follows. For $t = 2, \ldots, T-1$, the gradient of $\ell(\widetilde{\boldsymbol{x}}^{(j)})$ is:

$$\nabla\ell = \left[ \left( \frac{\partial\ell}{\partial\widetilde{\boldsymbol{x}}_1^{(j)}} \right)', \ldots, \left( \frac{\partial\ell}{\partial\widetilde{\boldsymbol{x}}_T^{(j)}} \right)' \right]',$$

$$\frac{\partial\ell}{\partial\widetilde{\boldsymbol{x}}_1^{(j)}} = \widetilde{\boldsymbol{C}}'^{(j)}(\widetilde{\boldsymbol{y}}_1^{(j)} - \widetilde{\boldsymbol{\lambda}}_1) - [\widetilde{\boldsymbol{Q}}_0^{(j)}]^{-1}(\widetilde{\boldsymbol{x}}_1^{(j)} - \widetilde{\boldsymbol{x}}_0^{(j)}) + \widetilde{\boldsymbol{A}}'^{(j)}[\widetilde{\boldsymbol{Q}}^{(j)}]^{-1}(\widetilde{\boldsymbol{x}}_2^{(j)} - \widetilde{\boldsymbol{A}}^{(j)}\widetilde{\boldsymbol{x}}_1^{(j)} - \widetilde{\boldsymbol{b}}^{(j)}),$$

$$\frac{\partial\ell}{\partial\widetilde{\boldsymbol{x}}_t^{(j)}} = \widetilde{\boldsymbol{C}}'^{(j)}(\widetilde{\boldsymbol{y}}_t^{(j)} - \widetilde{\boldsymbol{\lambda}}_t) - [\widetilde{\boldsymbol{Q}}^{(j)}]^{-1}(\widetilde{\boldsymbol{x}}_t^{(j)} - \widetilde{\boldsymbol{A}}^{(j)}\widetilde{\boldsymbol{x}}_{t-1}^{(j)} - \widetilde{\boldsymbol{b}}^{(j)})$$

$$+ \widetilde{\boldsymbol{A}}'^{(j)}[\widetilde{\boldsymbol{Q}}^{(j)}]^{-1}(\widetilde{\boldsymbol{x}}_{t+1}^{(j)} - \widetilde{\boldsymbol{A}}^{(j)}\widetilde{\boldsymbol{x}}_t^{(j)} - \widetilde{\boldsymbol{b}}^{(j)}),$$

$$\frac{\partial\ell}{\partial\widetilde{\boldsymbol{x}}_T^{(j)}} = \widetilde{\boldsymbol{C}}'^{(j)}(\widetilde{\boldsymbol{y}}_t^{(j)} - \widetilde{\boldsymbol{\lambda}}_T) - [\widetilde{\boldsymbol{Q}}^{(j)}]^{-1}(\widetilde{\boldsymbol{x}}_T^{(j)} - \widetilde{\boldsymbol{A}}^{(j)}\widetilde{\boldsymbol{x}}_{T-1}^{(j)} - \widetilde{\boldsymbol{b}}^{(j)}).$$

And the Hessian matrix is:

$$\nabla\nabla\ell = \begin{pmatrix} \frac{\partial^2\ell}{\partial\widetilde{\boldsymbol{x}}_1^{(j)}\partial\widetilde{\boldsymbol{x}}_1'^{(j)}} & \widetilde{\boldsymbol{A}}'^{(j)}[\widetilde{\boldsymbol{Q}}^{(j)}]^{-1} & \boldsymbol{0} & \cdots & \boldsymbol{0} \\ [\widetilde{\boldsymbol{Q}}^{(j)}]^{-1}\widetilde{\boldsymbol{A}}^{(j)} & \frac{\partial^2\ell}{\partial\widetilde{\boldsymbol{x}}_2^{(j)}\partial\widetilde{\boldsymbol{x}}_2'^{(j)}} & \widetilde{\boldsymbol{A}}'^{(j)}[\widetilde{\boldsymbol{Q}}^{(j)}]^{-1} & \cdots & \vdots \\ \boldsymbol{0} & [\widetilde{\boldsymbol{Q}}^{(j)}]^{-1}\widetilde{\boldsymbol{A}}^{(j)} & \frac{\partial^2\ell}{\partial\widetilde{\boldsymbol{x}}_3^{(j)}\partial\widetilde{\boldsymbol{x}}_3'^{(j)}} & \cdots & \vdots \\ \vdots & \vdots & \vdots & \ddots & \vdots \\ \boldsymbol{0} & \cdots & \cdots & \cdots & \frac{\partial^2\ell}{\partial\widetilde{\boldsymbol{x}}_T^{(j)}\partial\widetilde{\boldsymbol{x}}_T'^{(j)}} \end{pmatrix},$$

$$\frac{\partial^2\ell}{\partial\widetilde{\boldsymbol{x}}_1^{(j)}\partial\widetilde{\boldsymbol{x}}_1'^{(j)}} = -\widetilde{\boldsymbol{C}}'^{(j)}diag(\widetilde{\boldsymbol{\lambda}}_1)\widetilde{\boldsymbol{C}}^{(j)} - [\widetilde{\boldsymbol{Q}}_0^{(j)}]^{-1} - \widetilde{\boldsymbol{A}}'^{(j)}[\widetilde{\boldsymbol{Q}}^{(j)}]^{-1}\widetilde{\boldsymbol{A}}^{(j)},$$

$$\frac{\partial^2\ell}{\partial\widetilde{\boldsymbol{x}}_t^{(j)}\partial\widetilde{\boldsymbol{x}}_t'^{(j)}} = -\widetilde{\boldsymbol{C}}'^{(j)}diag(\widetilde{\boldsymbol{\lambda}}_t)\widetilde{\boldsymbol{C}}^{(j)} - [\widetilde{\boldsymbol{Q}}^{(j)}]^{-1} - \widetilde{\boldsymbol{A}}'^{(j)}[\widetilde{\boldsymbol{Q}}^{(j)}]^{-1}\widetilde{\boldsymbol{A}}^{(j)},$$

$$\frac{\partial^2\ell}{\partial\widetilde{\boldsymbol{x}}_T^{(j)}\partial\widetilde{\boldsymbol{x}}_T'^{(j)}} = -\widetilde{\boldsymbol{C}}'^{(j)}diag(\widetilde{\boldsymbol{\lambda}}_T)\widetilde{\boldsymbol{C}}^{(j)} - [\widetilde{\boldsymbol{Q}}^{(j)}]^{-1}.$$

Although we can conduct the Newton update efficiently in $\mathcal{O}(T)$, bad initial values may slow down the convergence. To facilitate convergence, we initialize the Newton update with a smoothing estimate by local Gaussian approximation. The forward filtering for a dynamic Poisson model has been previously described (Eden et al., 2004), and we use an additional backward pass to smooth (Rauch et al., 1965).

Let $\widetilde{\boldsymbol{x}}_{t|t-1}^{(j)} = E(\widetilde{\boldsymbol{x}}_t^{(j)}|\widetilde{\boldsymbol{y}}_1^{(j)}, \ldots, \widetilde{\boldsymbol{y}}_{t-1}^{(j)})$ and $\boldsymbol{V}_{t|t-1}^{(j)} = Var(\widetilde{\boldsymbol{x}}_t^{(j)}|\widetilde{\boldsymbol{y}}_1^{(j)}, \ldots, \widetilde{\boldsymbol{y}}_{t-1}^{(j)})$ be the mean and variance for the one-step prediction density, where $\widetilde{\boldsymbol{x}}_{t|t-1}^{(j)} = \widetilde{\boldsymbol{A}}^{(j)}\widetilde{\boldsymbol{x}}_{t-1|t-1}^{(j)} + \widetilde{\boldsymbol{b}}^{(j)}$ and $\boldsymbol{V}_{t|t-1}^{(j)} = \widetilde{\boldsymbol{A}}^{(j)}\boldsymbol{V}_{t-1|t-1}^{(j)}\widetilde{\boldsymbol{A}}'^{(j)} + \widetilde{\boldsymbol{Q}}^{(j)}$. Then, after we observe the data at time $t$, we can do a forward filtering step for the mean $\widetilde{\boldsymbol{x}}_{t|t}^{(j)} = E(\widetilde{\boldsymbol{x}}_t^{(j)}|\widetilde{\boldsymbol{y}}_1^{(j)}, \ldots, \widetilde{\boldsymbol{y}}_t^{(j)})$ and the variance $\boldsymbol{V}_{t|t}^{(j)} = Var(\widetilde{\boldsymbol{x}}_t^{(j)}|\widetilde{\boldsymbol{y}}_1^{(j)}, \ldots, \widetilde{\boldsymbol{y}}_t^{(j)})$, which are given by

$$\widetilde{\boldsymbol{x}}_{t|t}^{(j)} = \widetilde{\boldsymbol{x}}_{t|t-1}^{(j)} + \boldsymbol{V}_{t|t-1}^{(j)}[\widetilde{\boldsymbol{C}}'^{(j)}(\widetilde{\boldsymbol{y}}_t^{(j)} - \widetilde{\boldsymbol{\lambda}}_t)]_{\widetilde{\boldsymbol{x}}_t^{(j)} = \widetilde{\boldsymbol{x}}_{t|t-1}^{(j)}},$$

$$[\boldsymbol{V}_{t|t}^{(j)}]^{-1} = [\boldsymbol{V}_{t|t-1}^{(j)}]^{-1} + [\widetilde{\boldsymbol{C}}'^{(j)}diag(\widetilde{\boldsymbol{\lambda}}_t)\widetilde{\boldsymbol{C}}^{(j)} - [\widetilde{\boldsymbol{Q}}^{(j)}]^{-1}]_{\widetilde{\boldsymbol{x}}_t^{(j)} = \widetilde{\boldsymbol{x}}_{t|t-1}^{(j)}}.$$

Derivation of the filtering estimates can be found in (Eden et al., 2004), and we can further get the smoothing estimates directly by standard Rauch-Tung-Striebel smoother (Rauch et al., 1965).

The smoother estimates $\widetilde{\boldsymbol{x}}_{t|T}^{(j)}$ and $\boldsymbol{V}_{t|T}^{(j)}$ are updated as follows:

$$\widetilde{\boldsymbol{x}}_{t-1|T}^{(j)} = \widetilde{\boldsymbol{x}}_{t-1|t-1}^{(j)} + \boldsymbol{J}_{t-1}(\widetilde{\boldsymbol{x}}_{t|T}^{(j)} - \widetilde{\boldsymbol{x}}_{t|t-1}^{(j)}),$$

$$\boldsymbol{V}_{t-1|T}^{(j)} = \boldsymbol{V}_{t-1|t-1}^{(j)} + \boldsymbol{J}_{t-1}(\boldsymbol{V}_{t|T}^{(j)} - \boldsymbol{V}_{t|t-1}^{(j)})\boldsymbol{J}_{t-1}',$$

where $\boldsymbol{J}_{t-1} = \boldsymbol{V}_{t-1|t-1}^{(j)}\widetilde{\boldsymbol{A}}'^{(j)}[\boldsymbol{V}_{t|t-1}^{(j)}]^{-1}$.

**Update $\delta_i$ and $\boldsymbol{c}_i$**  We specify the prior for neuron-specific baseline $\delta_i$ as $\delta_i \sim \mathcal{N}(0,1)$ and we have assumed the loading $\boldsymbol{c}_i \sim \mathcal{N}(\boldsymbol{0}, \boldsymbol{I}_p)$. Then, from the matrix representation of DPFA in Equation (1), i.e., $\log \boldsymbol{\lambda}_i = \delta_i \boldsymbol{1}_T + \boldsymbol{\mu}^{(j)} + \boldsymbol{X}^{(j)}\boldsymbol{c}_i$, it is easy to see that given $\boldsymbol{\mu}^{(j)}$ and $\boldsymbol{X}^{(j)}$ are known, the update of $\delta_i$ and $\boldsymbol{c}_i$ is just a regular Bayesian Poisson regression problem. Thus, we can sample the full conditional distribution of $\delta_i$ and $\boldsymbol{c}_i$ by a Hamiltonian Monte Carlo (HMC, Duane et al. 1987) within the Gibbs sampler.

**Update parameters of latent state**  The parameters for linear dynamics are $h^{(j)}, g^{(j)}, \sigma^{2(j)}, \boldsymbol{A}^{(j)}$, $\boldsymbol{b}^{(j)}$ and $\boldsymbol{Q}^{(j)}$. To make the model identifiable, we simply assume $\boldsymbol{A}^{(j)} = diag(a_1^{(j)}, \ldots, a_p^{(j)})$ and $\boldsymbol{Q}^{(j)} = diag(q_1^{(j)}, \ldots, q_p^{(j)})$. Therefore, we can update $\boldsymbol{A}^{(j)}, \boldsymbol{b}^{(j)}$ and $\boldsymbol{Q}^{(j)}$ for each diagonal element separately, as the update in $h^{(j)}, g^{(j)}$ and $\sigma^{2(j)}$. Here, we update $h^{(j)}, g^{(j)}$ and $\sigma^{2(j)}$ as follows.

First, we specify the priors for $\sigma^{2(j)}$ following $IG\left(\nu_0/2, \nu_0\sigma_0^2/2\right)$ and $\left(g^{(j)}, h^{(j)}\right)' \sim \mathcal{N}(\boldsymbol{\tau}_0, \sigma^{2(j)}\boldsymbol{\Lambda}_0^{-1})$, with $\nu_0 = 1$, $\sigma_0 = 0.01$, $\boldsymbol{\tau}_0 = (0,1)'$ and $\boldsymbol{\Lambda}_0 = \boldsymbol{I}_2$. Here, the "IG" denotes the inverse-gamma distribution.

Denote $\boldsymbol{\mu}_{2:T}^{(j)} = \left(\mu_2^{(j)}, \ldots, \mu_T^{(j)}\right)'$ and $\widetilde{\boldsymbol{\mu}}_{1:(T-1)}^{(j)} = \left(\boldsymbol{1}_{T-1}, \boldsymbol{\mu}_{1:(T-1)}^{(j)}\right)$, with $\boldsymbol{\mu}_{1:(T-1)}^{(j)} = \left(\mu_1^{(j)}, \ldots, \mu_{T-1}^{(j)}\right)'$. The full conditional distributions for $\sigma^{2(j)}$ and $\left(g^{(j)}, h^{(j)}\right)'$ are:

$$\sigma^{2(j)}|\{\mu_t^{(j)}\}_{t=1}^T \sim IG\left(\frac{\nu_0 + T - 1}{2}, \frac{\nu_0\sigma_0^2 + \boldsymbol{\mu}_{2:T}'^{(j)}\boldsymbol{\mu}_{2:T}^{(j)} + \boldsymbol{\tau}_0'\boldsymbol{\Lambda}_0\boldsymbol{\tau}_0 - \boldsymbol{\tau}_n'\boldsymbol{\Lambda}_n\boldsymbol{\tau}_n}{2}\right),$$

$$\left(g^{(j)}, h^{(j)}\right)'|\{\mu_t^{(j)}\}_{t=1}^T \sim \mathcal{N}(\boldsymbol{\tau}_n, \sigma^{2(j)}\boldsymbol{\Lambda}_n^{-1}),$$

with $\boldsymbol{\Lambda}_n = \widetilde{\boldsymbol{\mu}}_{1:(T-1)}'^{(j)}\widetilde{\boldsymbol{\mu}}_{1:(T-1)}^{(j)} + \boldsymbol{\Lambda}_0$, and $\boldsymbol{\tau}_n = \boldsymbol{\Lambda}_n^{-1}\left(\widetilde{\boldsymbol{\mu}}_{1:(T-1)}'^{(j)}\boldsymbol{\mu}_{2:T}^{(j)} + \boldsymbol{\Lambda}_0\boldsymbol{\tau}_0\right)$.

For completeness, we also provide the update of latent state parameters when using the more parsimonious constraint, i.e. diagonal $\boldsymbol{X}'^{(j)}\boldsymbol{X}^{(j)}$. According to results from previous research (Krzanowski and Marriott, 1994b,a; Fokoué and Titterington, 2003), this constraint is one of the most parsimonious one, which only put constraints on $p(p-1)/2$ parameters. The constraint can also be implemented in MCMC by "unconstrained sampling-projection" procedure (Sen et al., 2018).

Without constraints, the sampling of $h^{(j)}, g^{(j)}$ and $\sigma^{2(j)}$ is the same as shown previously. The update of $\boldsymbol{A}^{(j)}, \boldsymbol{b}^{(j)}$ and $\boldsymbol{Q}^{(j)}$ is the standard multivariate Bayesian linear regression. Denote $\boldsymbol{X}_{2:T}^{(j)} = (\boldsymbol{x}_2^{(j)}, \ldots, \boldsymbol{x}_T^{(j)})'$ and $\widetilde{\boldsymbol{X}}_{1:(T-1)}^{(j)} = (\boldsymbol{1}_{T-1}, \boldsymbol{X}_{1:(T-1)}^{(j)})$, with $\boldsymbol{X}_{1:(T-1)}^{(j)} = (\boldsymbol{x}_1^{(j)}, \ldots, \boldsymbol{x}_{T-1}^{(j)})'$. Let us use the conjugate priors as following for $\boldsymbol{Q}^{(j)}, \boldsymbol{b}^{(j)}$ and $\boldsymbol{A}^{(j)}$:

$$\boldsymbol{Q}^{(j)} \sim \mathcal{W}^{-1}(\boldsymbol{\Psi}_0, \gamma_0),$$

$$vec((\boldsymbol{b}^{(j)}, \boldsymbol{A}'^{(j)})) \sim N(vec(\boldsymbol{T}_0), \boldsymbol{Q}^{(j)} \otimes \boldsymbol{\Gamma}_0^{-1}).$$

Here, the $\mathcal{W}^{-1}$ denotes the inverse-Wishart distribution. We can set the priors as $\boldsymbol{\Psi}_0 = 0.01\boldsymbol{I}_p$, $\gamma_0 = p+2$ and $\boldsymbol{T}_0 = (\boldsymbol{0}_p, \boldsymbol{I}_p)'$. Then, the full conditional distributions for $\boldsymbol{Q}^{(j)}$ and $vec((\boldsymbol{b}^{(j)}, \boldsymbol{A}^{(j)})')$ are:

$$\boldsymbol{Q}^{(j)}|\boldsymbol{X}^{(j)} \sim \mathcal{W}^{-1}(\boldsymbol{\Psi}_n, \gamma_n),$$

$$vec((\boldsymbol{b}^{(j)}, \boldsymbol{A}^{(j)})')|\boldsymbol{X}^{(j)} \sim N(vec(\boldsymbol{T}_n), \boldsymbol{Q}^{(j)} \otimes \boldsymbol{\Gamma}_n^{-1}),$$

with

$$\boldsymbol{\Psi}_n = \boldsymbol{\Psi}_0 + (\boldsymbol{X}_{2:T}^{(j)} - \widetilde{\boldsymbol{X}}_{1:(T-1)}^{(j)}\boldsymbol{T}_n)'(\boldsymbol{X}_{2:T}^{(j)} - \widetilde{\boldsymbol{X}}_{1:(T-1)}^{(j)}\boldsymbol{T}_n),$$
$$+ (\boldsymbol{T}_n - \boldsymbol{T}_0)'\boldsymbol{\Gamma}_0(\boldsymbol{T}_n - \boldsymbol{T}_0),$$
$$\gamma_n = \gamma_0 + T - 1,$$
$$\boldsymbol{T}_n = \boldsymbol{\Gamma}_n^{-1}(\widetilde{\boldsymbol{X}}_{1:(T-1)}'^{(j)}\boldsymbol{X}_{2:T}^{(j)} + \boldsymbol{\Gamma}_0\boldsymbol{T}_0),$$
$$\boldsymbol{\Gamma}_n = \widetilde{\boldsymbol{X}}_{1:(T-1)}'^{(j)}\widetilde{\boldsymbol{X}}_{1:(T-1)}^{(j)} + \boldsymbol{\Gamma}_0.$$

Before updating the cluster indices $z_i$, the sampling of $\{\boldsymbol{x}_t^{(j)}, \mu_t^{(j)}, \delta_i, \boldsymbol{c}_i, \boldsymbol{\theta}^{(j)}\}$ is repeated several times (5 times in both simulation and application for this paper). To save time, we can further allow the repetitions to be pre-stopped, when the training log-likelihood converges roughly.

**Update $z_i$**    To update the cluster assignments for each neuron $i$, we use a partition based algorithm for MFM, similarly as described in Miller and Harrison 2018.

Let $\mathcal{C}$ denote a partition of neurons, and $\mathcal{C}\backslash i$ denote the partition obtained by removing neuron $i$ from $\mathcal{C}$.

1. Initialize $\mathcal{C}$ and $\{\boldsymbol{\theta}^{(c)} : c \in \mathcal{C}\}$ (e.g. one cluster).
2. Repeat the following steps $G$ times to obtain $G$ samples. For $i = 1, \ldots, N$: remove neuron $i$ from $\mathcal{C}$ and place it:
    (a) in $c \in \mathcal{C}\backslash i$ with probability $\propto (|c| + \gamma)M_{\boldsymbol{\theta}^{(c)}}(\boldsymbol{y}_i)$, where $\gamma$ is defined in the MFM model in the main text (Equation 2) and $M_{\boldsymbol{\theta}^{(c)}}(\boldsymbol{y}_i)$ denotes the marginal likelihood of neuron $i$ in cluster $c$, when integrating the loading $\boldsymbol{c}_i$ out (Equation (3)). The marginal likelihood is approximated by using Poisson-Gamma conjugacy.
    (b) in a new cluster $c^*$ with probability $\propto \gamma\frac{V_n(t+1)}{V_n(t)}M_{\boldsymbol{\theta}^{(c^*)}}(\boldsymbol{y}_i)$, where $t$ is the number of partitions obtained by removing the neuron $i$ and $V_n(t) = \sum_{j=1}^{\infty} \frac{j_{(t)}}{(\gamma j)^{(n)}}f_k(j)$, with $x^{(m)} = x(x+1)\cdots(x+m-1)$, $x_{(m)} = x(x-1)\cdots(x-m+1)$, $x^{(0)} = 1$ and $x_{(0)} = 1$.

The update is an adaptation of partition-based algorithm for DPM (Neal, 2000), but with two substitutions: 1) replace $|c_i|$ by $|c_i| + \gamma$ and 2) replace $\alpha$ by $\gamma V_n(t+1)/V_n(t)$. See more details and discussions in (Miller and Harrison, 2018). When evaluating the likelihood, we marginalize the cluster-independent loading $\boldsymbol{c}_i$ out. This is necessary for the high dimensional situation, otherwise the chain will stop moving.

One issue with incremental Gibbs samplers such as Algorithm 3 and 8 in Neal (2000), when applied to DPM, is that mixing can be somewhat slow. To further improve the mixing, we may intersperse the "split-merge" Metropolis-Hasting updates (Jain and Neal, 2007, 2004) between Gibbs sweeps, as in (Miller and Harrison, 2018).

## A.2    Sample Latent Vectors Using Pólya-Gamma Augmentation and Metropolis-Hastings Algorithm

In this section, we provide details of sampling algorithms to draw the latent states $\boldsymbol{X}^{(j)}$ and population baseline $\boldsymbol{\mu}^{(j)}$ from the exact posterior of the model. In order to explain the algorithm much clearer, we just focus on a given cluster index, and thus in this section we suspend the superscript $(j)$ as the cluster index in our notations. We present the sampling idea with two major parts as described below.

**Pólya-Gamma Augmentation**    Although the mixDPFA model does not directly follow the PG augmentation scheme (Polson et al., 2013), we can approximate the Poisson distribution by a negative binomial (NB) distribution, i.e., $\lim_{r\to\infty} \text{NB}(r, \sigma(\psi - \log r)) = \text{Poisson}(e^\psi)$, where $\sigma(\psi) = e^\psi/(1 + e^\psi)$ and $\text{NB}(r, p)$ denotes the NB distribution with $rp/(1 - p)$ as its expectation. We approximate the proposed DPFA model using a NB, then we can instead sample a mixture of NB factor analyzers using the PG scheme (Windle et al., 2013). Further, we use the forward-filtering-backward-sampling (FFBS) algorithm (Carter and Kohn, 1994; Frühwirth-Schnatter, 1994) to update

the latent states $X$ and population baseline $\mu$. These two steps are summarized in the step 1 and step 2 of Algorithm 1 for updating $\widetilde{X} = (\mu, X)$.

**Metropolis-Hastings Step**   Next, we use the samples of $\widetilde{X}$ yielded from FFBS algorithm as a proposal, where we employ a Metropolis-Hastings (MH) step to reject or accept the proposal. In this step, the dispersion parameter $r$ in NB distribution becomes a tuning parameter, to balance acceptance rate and autocorrelation in MH. When $r$ is large, the approximation to Poisson observation is accurate and the MH performs similar to the Gibbs sampler. Here, we allow neurons at different time points to have unique tuning parameters $r_{it}$. The MH step is summarized in step 3 of Algorithm 1.

We have further implemented Algorithm 1 to fit DPFA model for cluster 1 data in Fig. 2 and the results have been presented in Fig. 1. We set the tuning parameters for each neuron at all time points as $r_{it} = 10$ , to tune the acceptance rate around 0.4. When implementing the PG-MH algorithm in DPFA, the $\omega_{it}$ (i.e., PG random variable) was sampled using `pgdraw` function in R package `pgdraw`, which is called from our MATLAB code. Although we can program Algorithm 1 without calling `pgdraw` function from R to save some time, to sample the PG random variable is often much computationally expensive in comparison to sample from the Gaussian random variable. When fitting the simulated data with $N = 5$, $T = 1000$ and $p = 2$ (cluster 1 data in Fig. 2) using a 3.40 GHz processor with 16 GB of RAM, it takes 131.64s for PG-MH to draw 1000 posterior samples, while the proposed approximation method in the main context of the paper takes 83.86s. Although the proposed method takes the approximation to the full conditional of the latent state $X$ and population baseline $\mu$, the results are similar as to sample directly from the exact full conditional. Table 1 shows the similarities, defined by cosine function, of fitted $\mu$ and $X = (X_{*1}, X_{*2})$ between the PG-MH and the Gaussian approximation for full conditional, where $X_{*l}$ denotes the $l$-th latent vectors (i.e. the $l$-th column of $X$). Here, $\hat{\xi}_{\text{PGMH}}$ denotes the average from iteration 5000 to 10,000 for chain 1 of PG-MH (Fig. 1), $\hat{\xi}_{\mathcal{N}}$ denotes the average from iteration 5000 to 10,000 for the chain in Fig. 2 of the Gaussian approximation, and $\xi_{\text{true}}$ denotes the ground truth. In each column, $\xi$ is replaced by $\mu$, $X_{*1}$ and $X_{*2}$ respectively. The posterior means for these two method are close to each other, i.e. $\cos(\hat{\xi}_{\text{PGMH}}, \hat{\xi}_{\mathcal{N}}) \approx 1$.

### A.3  Supplementary Results for Neuropixels Application

This section shows supplementary results when applying the proposed model to the Neuropixels dataset (4 Multi-region neural spike recordings). Here, we show 1) results sorted by maximum a posteriori probability (MAP) estimates and 2) clustering results in another independent chain (Fig. 2).

**Algorithm 1:** Pólya-Gamma-Metropolis-Hastings Algorithm (PG-MH) for Poisson Dynamic Model

Recall the DPFA model using the notations defined in A.1:

$$y_{it} \sim Poi(\lambda_{it}),$$
$$\log \lambda_{it} = \delta_i + \widetilde{c}_i' \widetilde{x}_t,$$

for $i = 1, \ldots, n$ and $t = 1, \ldots, T$. Here, $\widetilde{c}_i = (1, c_i')'$ and $\widetilde{x}_t = (\mu_t, x_t')'$. $\widetilde{x}_t$ follows linear dynamics $\widetilde{x}_{t+1} | \widetilde{x}_t \sim \mathcal{N}(\widetilde{A}\widetilde{x}_t + \widetilde{b}, \widetilde{Q})$. Denote the prior as $\widetilde{x}_1 \sim \mathcal{N}(m_0, V_0)$. Given the sample from the $(G-1)$-th iteration $\widetilde{x}_t^{(G-1)}$ and $U = \{\widetilde{c}_i, \delta_i, \widetilde{A}, \widetilde{b}, \widetilde{Q}\}$.

**1.** sample $\omega_{it}$ from PG distribution and calculate $\hat{y}_{it}$, which follows $\mathcal{N}(\widetilde{c}_i' \widetilde{x}_t^{(G-1)}, \omega_{it}^{-1})$
**for** $t = 1, \ldots, T$ **do**
    **for** $i = 1, \ldots, n$ **do**
        sample $\omega_{it} \sim P_{PG}(r_{it} + y_{it}, \delta_i + \widetilde{c}_i' \widetilde{x}_t^{(G-1)} - \log r_{it})$
        $\kappa_{it} = (y_{it} - r_{it})/2 + \omega_{it}(\log r_{it} - \delta_i)$
        $\hat{y}_{it} = \omega_{it}^{-1} \kappa_{it}$
    **end**
**end**

**2.** Forward-filtering-backward-sampling (FFBS) for $\widetilde{X}$
Denote $\hat{y}_t = (\hat{y}_{1t}, \ldots, \hat{y}_{Nt})'$, $\Omega_t = Diag([\omega_{1t}, \ldots, \omega_{Nt}])$ and $\widetilde{C} = (\widetilde{c}_1, \ldots, \widetilde{c}_N)'$
**for** $t = 1, \ldots, T$ **do**
    $m_{t|t-1} = \widetilde{A}m_{t-1} + \widetilde{b}$
    $V_{t|t-1} = \widetilde{A}V_{t-1}\widetilde{A}' + \widetilde{Q}$
    $K_t = V_{t|t-1}\widetilde{C}'(\widetilde{C}V_{t|t-1}\widetilde{C}' + \Omega_t^{-1})^{-1}$
    $m_t = m_{t|t-1} + K_t(\hat{y}_t - \widetilde{C}m_{t|t-1})$
    $V_t = (I - K_t\widetilde{C})V_{t|t-1}$
**end**
sample $\widetilde{x}_T^* \sim \mathcal{N}(m_T, V_T)$
**for** $t = T-1, \ldots, 1$ **do**
    $J_t = V_t\widetilde{A}'(\widetilde{A}V_t\widetilde{A}' + \widetilde{Q})^{-1}$
    $m_t^* = m_t + J_t(x_{t+1}^* - \widetilde{A}m_t - \widetilde{b})$
    $V_t^* = (I - J_t\widetilde{A})V_t$
    sample $x_t^* \sim \mathcal{N}(m_t^*, V_t^*)$
**end**

**3.** Accept or reject the proposal $\widetilde{X}^*$
compute the acceptance ratio

$$\zeta = \frac{\pi(\widetilde{X}^* | \{y_i\}_{i=1}^n, U)}{\pi(\widetilde{X}^{(G-1)} | \{y_i\}_{i=1}^n, U)} \frac{q(\widetilde{X}^{(G-1)} | \widetilde{X}^*, U)}{q(\widetilde{X}^* | \widetilde{X}^{(G-1)}, U)}$$
$$= \frac{P(\{y_i\}_{i=1}^n | \widetilde{X}^*)}{P(\{y_i\}_{i=1}^n | \widetilde{X}^{(G-1)})} \frac{NB(\{y_i\}_{i=1}^n | \widetilde{X}^{(G-1)}, R)}{NB(\{y_i\}_{i=1}^n | \widetilde{X}^*, R)},$$

where $R = \{r_{it}\}$ is matrix for dispersion parameters for each neuron at all time points. $P(\cdot)$ denotes the Poisson likelihood, and $NB(\cdot)$ denotes the negative binomial likelihood. Accept the proposal $\widetilde{X}^*$ with probability $\min(1, \zeta)$.

Table 1: PG-MH vs. Gaussian approximation for $\boldsymbol{\mu}$ and $\boldsymbol{X}$

| | $\boldsymbol{\mu}$ | $\boldsymbol{X}_{*1}$ | $\boldsymbol{X}_{*2}$ |
|---|---|---|---|
| $\cos\left(\hat{\boldsymbol{\xi}}_{\text{PGMH}}, \boldsymbol{\xi}_{\text{true}}\right)$ | 0.9724 | 0.7663 | 0.9728 |
| $\cos\left(\hat{\boldsymbol{\xi}}_{\mathcal{N}}, \boldsymbol{\xi}_{\text{true}}\right)$ | 0.9680 | 0.7443 | 0.9699 |
| $\cos\left(\hat{\boldsymbol{\xi}}_{\text{PGMH}}, \hat{\boldsymbol{\xi}}_{\mathcal{N}}\right)$ | 0.9435 | 0.9664 | 0.9959 |

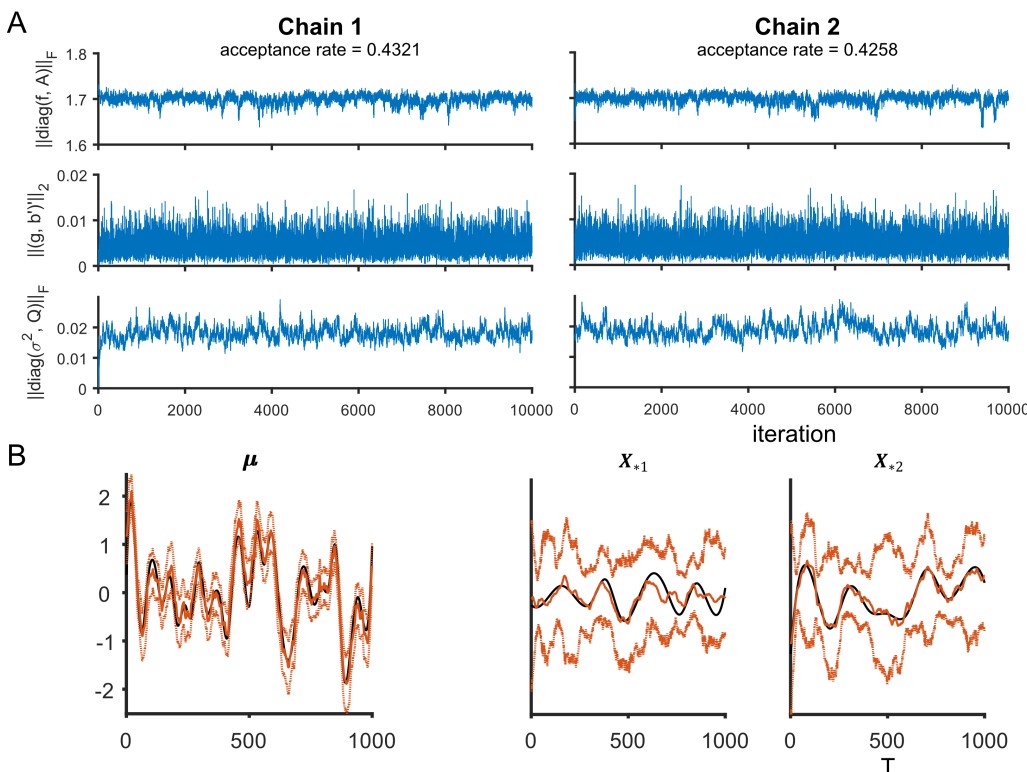

Figure 1: **PG-MH for DPFA** Two independent chains (10,000 iterations) for cluster 1 data in Figure 2. The full conditional distributions of $\boldsymbol{\mu}$ and $\boldsymbol{X}$ are sampled by PG-MH algorithm (Algorithm 1).**A.** Traceplot of Frobenius norms of linear dynamics. The acceptance rates in subtitle are for sampling $\boldsymbol{\mu}$ and $\boldsymbol{X}$ (PG-MH step). **B.** The true (black) and fitted (colored) population baseline and latent factor. Use samples from iteration 5000 to 10,000 in chain 1, the solid orange lines show averages and the dashed lines show 95% HPD intervals.

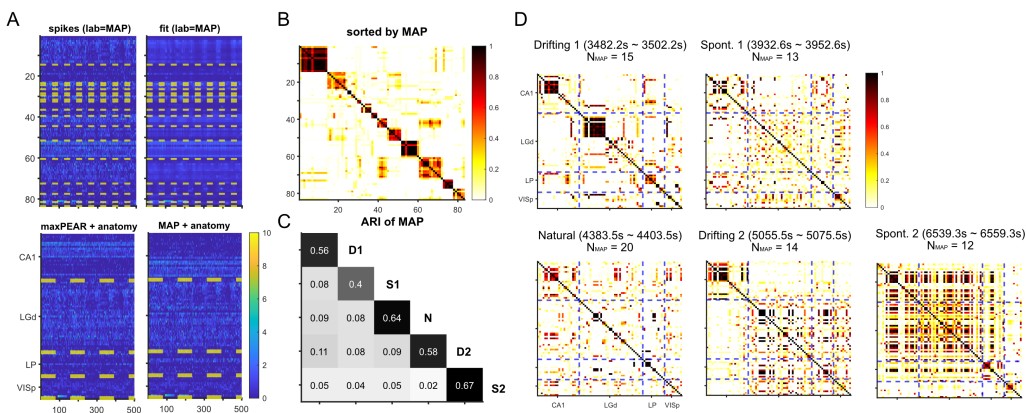

Figure 2: **Supplementary Results for Neuropixels Application A.** The first row shows the spike counts and fitted mean firing rate, sorted according to the MAP label estimates. The second row sort the spikes further according the anatomical sites, which shows clustering structure within each region. **B.** The posterior similarity matrix sorted by MAP estimates. **C.** The ARI of MAP estimates. The diagonal is ARI between 2 chains, and the off-diagonal is mean ARI of MAP for 4 combinations. **D.** Posterior similarity matrices for the second chains. Neurons are sorted as in the first panel of Fig. 4E