# OpenReview forum: "Bayesian Clustering of Neural Spiking Activity Using a Mixture of Dynamic Poisson Factor Analyzers"
_NeurIPS.cc/2022/Conference — NeurIPS 2022 Accept_

### Official Review · Reviewer_WZGP · 2022-06-29

**Rating:** 6
**Confidence:** 5
**Soundness:** 4 excellent
**Presentation:** 3 good
**Contribution:** 4 excellent

**Summary:**

The authors develop a novel model based on a mixture of dynamic Poisson factor analyzers to cluster neurons in large-scale neural recordings. They treat the number of clusters treated as an unknown parameter and propose a novel MCMC algorithm to efficiently sample its posterior distribution. They show in a simulation that their model can accurately recover the true clustering and latent states, and then explore the results of their method on a recording from the Allen Institute Visual Coding Neuropixels dataset.

**Questions:**

The model is very interesting and well constructed. However, it is a bit hard to grasp, and I feel like the section 2. could be a little more clear. I find that the graph in Fig. 1, C is not super informative, and could include more information, such as the priors and the additional parameters. H is also never introduced before it is mentioned on line 105. I understand that it corresponds to h, but in this case shouldn't there be an additional line in Equation 1 for mu and H?

I would have also liked to gain more intuition on why you chose the different priors. For example, I understand the use of a Geometric distribution as the prior on k, and that the mean of this prior should increase when we want to get more clusters, but an experiment showing the different results and recommended parameter with different datasets, probes, tasks, or number of brain region recording would help a lot.

You propose to use cross-validation for choosing p. I assume you optimize over the log-likelihood. Won't the log-likelihood necessarily increase as you use higher values for p? What criteria is used?

The latent states are chosen to evolve linearly over time with Gaussian noise. This makes sense when the animal is not doing any tasks. What happens when the animal change state, receives a stimulus or start doing a task? It would be great to add a visualization of the latent states throughout trials.

In section 2.3, you mention that "Though we may evaluate it by a Laplace approximation, but iterating over all potential clusters for each neuron is computationally intensive. To make faster clustering, we approximate the marginal likelihood by utilizing a Poisson-Gamma conjugacy." and "Another possible idea is to approximate the log-likelihood by second-order polynomials, with coefficients determined by Chebyshev polynomial approximation [Keeley et al., 2019]. However, we find that this approximation doesn’t work well in practice when spike counts have a wide range.". Could you show results and comparisons, in both speed and accuracy for these three approaches?

The main flaw of the model, to me, is that "Low firing rates tend to cause confusions in clustering". Low-firing rate neurons can encode crucial information in the brain, and artificially removing neurons with rates < 1Hz in the experiments does not seem to have any biological justification, but simply a way to get nicer results ass the firing rates will be more uniform. Some neurons can also one strongly associated to a state of the animal and have overall low firing rate as they fire strongly during specific tasks.
Could this be due to the choice of prior on lambda (and more specifically on delta/mu?)? The prior on delta is not specified (unless I miss it), but this could maybe help and solve this issue?
Moreover, the results with low-firing rate units are not even included in the paper. It is essential to include them (at least in the appendix) to have a sense of how it contaminates the clustering results.
What proportion of the neurons are discarded when keeping only neurons with FR>1Hz?

The low firing rate neurons can also be due to artifacts of spike sorting. It is especially true in Neuropixels datasets, where neurons can be split and thus have their firing rate cut in half (or more). There are many quality metrics developed for the output of spike sorting. (Hill et al. (2011), Harris et al. (2001), Schmitzer-Torbert and Redish. J Neurophy (2004), Chung et al. (2017)...). I think using such metrics to discard "bad" units would be much more appropriate (as is done in Reproducibility of in-vivo electrophysiological measurements in mice, International Brain Laboratory (2022) for example).
it would also be interesting to look at how the clusters are impacted by spike sorting artifacts. If you split a neuron in half, are the two half clustered together? How are the clusters impacted by electrode drift? You need to make sure the clustering algorithm is robust to these sources of noise.

What exactly is the "anatomical cluster model" mentioned in the result section. Are you running your clustering model on each region independently, using different values for p? (p=[1,11,5,3]). How do you choose the number of clusters per region? Do you use Cross-validation on each region? If you cluster neurons from each region independently, are the neurons clustered together also clustered together when clustering all neurons together? This would be a very interesting experiment to show in the paper.

Finally, MCMC models can be computationally expensive, and your model requires (5-fold) cross-validation. How does it scale with the number of neurons and length of the recording?

**Limitations:**

I don't see any potential negative societal impact of this work.

**Strengths And Weaknesses:**

The strength of the paper lies in the proposed model, that is novel and very well-thought. It is technically sound and well constructed. However, it is a complicated model and the section 2 of the paper could be a little more clear and give more intuition behind the choice of priors and parameters. The results on simulated data are very compelling, but I would have liked to see more experiments on real data (for example, how to choose the parameter of the geometric prior on k depending on the type of probe, or the number of anatomical regions) to gain more intuition. The main weakness of the model lies in the fact that "Low firing rates tend to cause confusions in clustering" and are excluded from the data in the analysis. This does not seem very biologically relevant.

---

> ### Author Response · Authors · 2022-07-29
> **Response**
>
> Thanks to the reviewer for their positive comments that the proposed model “is novel and very well-thought”. We have now tried to clarify several technical details..
>
> > However, it is a bit hard to grasp, and I feel like the section 2 could be a little more clear.
>
> **H** contains all the prior for parameters theta for cluster j. The details of prior H for all DPFA parameters can be found in each subsection of A.1 MCMC updates. H is included in (Fig 1C) to indicate the prior, however the other parameters that describe the evolution of the latent states are excluded for clarity.
>
> > I would have also liked to gain more intuition on why you chose the different priors.
>
> For the choice of k in the prior for the number of clusters, we now point out that for long recording times the likelihood will dominate the prior, so that the number of clusters is largely independent of the choice of prior.
>
> > You propose to use cross-validation for choosing p...
>
> When choosing p, we use the held-out log-likelihood (cross-validation). When we use very large p (larger than the true p), we will overfit the data. so the held-out log-likelihood will begin to drop as p increases beyond the ground truth p.
>
> > What happens when the animal change state, receives a stimulus or start doing a task?
>
> For modeling state changes, such as the start of a task, it’s right that the linear model may not be the best description of the state. Models such as the switching-LDS (SLDS, Fox, 2009 & Murphy, 2012) may be able to describe these effects, but here we focus on clustering neurons.
>
> > It would be great to add a visualization of the latent states throughout trials.
>
> When sampling the cluster labels, unlike with deterministic algorithms, it’s somewhat difficult to visualize, since every sample changes the cluster memberships. However, this is something we are aiming to follow up with for future work, where the relationship between latent states and experimental variables can be explored in more detail.
>
> > Could you show results and comparisons, in both speed and accuracy for these three approaches?
>
> Comparing the speed and accuracy for the approximations of the marginal likelihood for cluster membership (Laplace approximation, Gamma approximation, and 2nd-order polynomial approximation) is a nice idea, but somewhat beyond the scope of the paper at the moment. At least in our simulations, when evaluating the marginal likelihood by the 2nd-order polynomial [Keeley et al., 2019], it doesn't recover the ground truth cluster, no matter how we choose the approximation intervals (e.g. 1 interval for each neuron as suggested in their paper, or different intervals for both different neurons and time points). Both Laplace approximation and the Gamma approximation used in our paper can recover the ground truth cluster. For more detail about the Gamma approximation, we refer readers to (El-Sayyad, 1973 & Chan and Vasconcelos, 2009).
>
> > The main flaw of the model, to me, is that "Low firing rates tend to cause confusions in clustering".
>
> Thanks to the reviewer for pointing out the issue with low firing rate neurons. We now clarify to say that “Low firing rates and short recording lengths tend to cause confusions in clustering” because of weak information. We can handle the low firing rate case without a lot of confusion, if we fit the model with a long recording length T. We view the “confusion” as a strength – our model can accurately reflect the uncertainty in cluster membership for neurons with little information. Just giving a single point estimate of clustering doesn’t make a lot of sense, when the observation is not rich.
>
> Here we only include neurons with firing rates >1Hz, and have now added explicitly that this was 72% of the neurons in the recording. The experimental results are shown here mostly as a proof-of-concept that this approach gives potentially interesting clusters, but we hope to follow up with a more extensive analysis that will evaluate the effects of firing rates, recording length, bin size, spike sorting errors, task, and brain state more comprehensively.
>
> > What exactly is the "anatomical cluster model" mentioned in the result section?
>
> We have now modified the text to clarify that with this model the clusters are defined by the anatomical regions. The goal is to compare learned clusters, with a typical experimental definition of subpopulations just based on anatomy. For each region, we use cross-validation to select p separately. However, we didn’t do clustering for each region: neurons in each region are fitted by a single population DPFA.
>
> > Finally, MCMC models can be computationally expensive,...
>
> As far as computation, cross-validation is not very cumbersome, since we select p by checking the trace of held-out log-likelihood for short chains. The method is (approximately) linear in the length of the recording.

---

> > ### Comment · Reviewer_WZGP · 2022-08-03
> > **Re: Reviewer WZGP**
> >
> > I thank the authors for their answers and clarifications.
> >
> > The answers about the clarity of the paper, the choice of prior, the use of cross-validation for selecting p, and the definition of "anatomical cluster model" are satisfactory. However, I my major comments are still mostly unresolved and I am thus not willing to increase my score. I think the direction pursued by the authors of the paper is very interesting but the model still needs more validation.
> >
> > About the change of state (related to tasks): I understand that the authors focus on clustering neurons, but many neurons's firing rates will be modulated by these states, and the animals do perform tasks in the recordings considered in this paper. I do think that it would be necessary to consider models such as the switching-LDS, or at least evaluate if the clustering is consistent over trials of the recordings, how the different states impact the clustering. This is related to my comment about robustness to spike sorting artifacts. If you split a neuron in half, are both halves clustered together?
> > Moreover, does using these clusters give you higher accuracy on simple decoding tasks? What information do the clusters contain about the trials?
> >
> > About the fact that "Low firing rates tend to cause confusions in clustering":
> > 72% of the neurons in the recording is a lot, and discarding neurons based on firing rates is a huge assumption about the data. There is a priori no reason for a neuron to have high firing rate to be relevant. Moreover, have you checked if some neurons have overall low firing rate but have high firing rate for short periods of time (in which case they can contain important information about movement for example)?
> >
> > About the "speed and accuracy" of the model. I understand that the model is (approximately) linear in the length of the recording, which is great! Would it allow to perform online clustering? (faster than time)

---

> > > ### Author Response · Authors · 2022-08-05
> > > **Thank you**
> > >
> > > Thanks for these additional comments.
> > >
> > > A comparison with switching models, looking at decoding by cluster, and examining the latent states in more detail would all be interesting directions for future work.
> > >
> > > For low firing rates - we *include* 72% of neurons in the analysis. We've now tried to clarify in the text - it is not difficult or problematic to include low firing rate neurons, but we exclude them here for clarity.

---

### Official Review · Reviewer_dDsp · 2022-07-09

**Rating:** 4
**Confidence:** 5
**Soundness:** 4 excellent
**Presentation:** 3 good
**Contribution:** 2 fair

**Summary:**

This manuscript proposes a new approach called "Dynamic Poisson Factor Analyzers" (DPFA) that models neural spike trains.  This methodology is developed to split individual neurals into distinct clusters, based upon the hypothesis that neural spike trains from each neural cluster into distinct populations.  This approach uses fast approximations and modern MCMC approaches to implement an inference scheme, which is applied to synthetic data and high-dimensional real datasets.  Results are compared to essentially a PLDS model (no clustering) and a clustering model based upon anatomical location.

**Questions:**

What scientific question will this methodology help answer?

In what real-world datasets could this method significantly outperform the baselines?

Assuming discrete clustering of neurons is a strong assumption.  To my knowledge, most neurons participate in multiple subnetworks of the brain.  Why are the authors considering an explicit clustering of neurons rather than an admixture, which may be more biologically realistic?

**Limitations:**

No concerns.

**Strengths And Weaknesses:**

The biggest strength and contribution of this manuscript is the inference algorithm.  The inference algorithm seems very well-considered and explored, using fast approximations in the form of well-design Laplacian approximations and approximate conjugate distributions.  Combining these tricks with a Mixture of Finite Mixture (MFM) approach, rather than a Dirichlet Process, seems to mix fast and provide robust inference.  This approach seems fairly original and significant, although I would appreciate additional details and discussion on the efficacy and reliability of individual approximations (e.g., how close is the gamma-poisson distribution to the lognormal?).

The model itself is a reasonable model, but does not seem especially novel or significant given the existence of MixPLDS.  There are changes and advances, but they seem relatively minor.  It would be helpful if the authors could highlight their particular advances more clearly, and explicit discuss how this improves the model.

One of the areas that would benefit from improvement is clarity on how this relates to a clear scientific challenge.  How the neurons cluster is potentially interesting, but I believe that the manuscript would be greatly improved by focusing on a clear scientific question that this methodology can help answer.

The results on real data, though, are one of the biggest weaknesses.  In the real data, the proposed method does not fit the data better based on log likelihood than a clustering based on known anatomy, and is barely better than not using a clustering approach at all.  It is difficult to motivate using this approach based on those results for me, especially as other competing approaches such as deep RNNs (e.g., LFADS) do tend to fit the data better.

---

> ### Author Response · Authors · 2022-07-29
> **Response**
>
> Thanks to the reviewer for their positive comments that “this approach seems fairly original and significant”. Here, we clarified some specific points…
>
> > I would appreciate additional details and discussion on the efficacy and reliability of individual approximations...
>
> As suggested, we have now added some additional detail and discussion on the approximations here. Using Poisson-gamma conjugacy (i.e. approximate lognormal by Gamma) in approximating the marginal likelihood for the cluster labels works well in our case. We find that when comparing it with Laplace approximation, the clustering results are essentially the same and recover the ground truth in simulation. We refer readers to El-Sayyad, 1973 & Chan and Vasconcelos, 2009 for more detailed results about efficacy and reliability.
>
> > The model itself is a reasonable model, but does not seem especially novel or significant given the existence of MixPLDS.
>
> In comparing our model (mixDPFA) to mixPLDS, it’s true that the model setting is quite similar. The main contribution is the inference. We try to emphasize the inference limitations of mixPLDS in the introduction: “1) it requires we predetermine the number of clusters, and 2) the clustering results are often sensitive to the initial cluster assignment.” The main contribution of this paper is providing solutions to these two inference problems. We also directly evaluate the accuracy of the Laplace approximation for the latent state by comparing the results to sampling of the exact posterior with Poly-Gamma augmentation (Fig 5 and Table 1). Additionally, here we have explicitly addressed model identifiability. For clustering, the constraints on $\sum_{t=1}^{T}\mu_t^{(j)} = 0$ and $\sum_{t=1}^{T}\mathbf{x}^{(j)}_t = \mathbf{0}$ are essential, since inappropriate constraints will change the clustering results significantly.
>
> > One of the areas that would benefit from improvement is clarity on how this relates to a clear scientific challenge.
>
> We have also tried to emphasize the scientific questions, as pointed out by the reviewer. The main scientific questions are "How is population neural activity structured in clusters? And what low-d latent descriptions are there for these clusters?" Beyond that, it depends on the specific data being fit. The clusters are interesting, largely because they represent a functional grouping that disagrees with the usual groupings based on anatomy or cell type. The distinct low-d latent states that describe each cluster could potentially have a more direct relationship to circuit function or perception/behavior than a single global low-d description of the entire population.
>
> > The results on real data, though, are one of the biggest weaknesses.
>
> It’s true that the improvement in held-out log likelihood is somewhat unremarkable. However, our model uses many fewer parameters, and the main purpose is to do clustering. Our model can extract more structure (i.e. clustering structure), with improvement in terms of model fitting as the “byproduct” (at least not worse than current methods).
>
> > In what real-world datasets could this method significantly outperform the baselines?
>
> In the introduction, we mention how this approach builds on mixtures of (Gaussian) factor analyzers (MFA), and the situations where mixDPFA outperforms other models for neural data would be somewhat similar. Namely, when the data is “globally nonlinear” and well described by combining a number of local factor analyzers. As large scale neuroscience records from many brain regions at once, this approach may be useful for describing disparate brain regions.

---

> > ### Comment · Reviewer_dDsp · 2022-08-04
> > **Re: Response**
> >
> > Thank you for clarifying your contributions relative to MixPLDS, especially highlighting the identifiability.  That is helpful context.  If you have not revised your manuscript yet to make this contribution clearer in the manuscript, I would suggest doing that.
> >
> > > Beyond that, it depends on the specific data being fit. The clusters are interesting, largely because they represent a functional grouping that disagrees with the usual groupings based on anatomy or cell type.
> >
> > I agree that the clustering could be interesting way to explore optimal functional groupings, and I see the direct comparison to anatomy.  However, I don't see a comparison to cell type--if that is in the manuscript, can you please point me to it?
> >
> > > The distinct low-d latent states that describe each cluster could potentially have a more direct relationship to circuit function or perception/behavior than a single global low-d description of the entire population.
> >
> > I agree that the clusters _could_ have a more direct relationship to circuit function, which is one of the hopes of mixture/admixture models used in the brain.  The scientific utility of this manuscript would be stronger if there was clear evidence that this was the case, which I feel is missing in the current manuscript.
> >
> > > the main purpose is to do clustering. Our model can extract more structure (i.e. clustering structure), with improvement in terms of model fitting as the “byproduct” (at least not worse than current methods).
> >
> > The clustering is interesting, but without evidence linking it to circuit function or perception/behavior (as the authors list above), this just seems a bit speculative without clear and strong predictive improvements.
> >
> > The neuropixels data has associated visual stimuli.  Would it be feasible to link the cluster responses to parts of the visual stimuli (maybe CCA between each cluster's neurons and the stimuli pattern)?  That would provide a clearer biological link and provide me with stronger evidence on the utility of the method.
> >
> > I am remain interested on the author's view on a clustering vs admixture assumption. Why is clustering the correct assumption?

---

> > > ### Author Response · Authors · 2022-08-05
> > > **Response**
> > >
> > > Thanks for these additional questions.
> > >
> > > We do not currently include a comparison with cell types or a comparison between clusters and stimulus responses. However, these, along with more detailed look at the latent states, would be interesting directions for future work.
> > >
> > > From what we understand about the admixture assumption, it is most applicable in cases where each datapoint consists of multiple parts (e.g. genetic data). We have not seen it previously used in computational neuroscience, but, since the admixture assumption is potentially more flexible than the mixture model, maybe there is some useful application. Are there any specific references that the reviewer has in mind?

---

### Official Review · Reviewer_QfU5 · 2022-07-12

**Rating:** 7
**Confidence:** 3
**Soundness:** 4 excellent
**Presentation:** 3 good
**Contribution:** 4 excellent

**Summary:**

The authors propose a model to identify distinct functional clusters of neurons in neural population activity without an an a priori determined number of clusters. Each cluster is characterized by a latent LDS and mapped through an exponential nonlinearity to poisson rates. The prior over number of clusters and cluster assignment is computed efficiently borrowing from recent work on a mixture of finite mixture (MFM) model. The authors use a bespoke procedure for using MCMC to evaluate posteriors for their model, which involves a laplace approximation to estimate the population states, and a poisson-gamma conjugacy to marginalize over of the latent loadings $c_i$ to evaluate the probability of a neuron belonging to a given cluster. The authors show results on simulated data where the true number of clusters is known, and then evaluate the model on neuropixel recordings. They show on the neuropixel data that the clusters identified using their method outperform anatomical clustering or a single cluster on held-out data. This work is an important step in identifying functional groups of neurons in large multi-region neural population recordings.

**Questions:**

 Some more detailed clarification on the inference would be helpful. Could you expand on how the laplace approximation is being used in 122-124? More organization in the introductory part of the inference section would I think go a long way. An equation showing the P(x|..) being approximated by Laplace would be helpful. Also it appears there is a typo in the third line of the laplace approximation in the appendix (floating closed parentheses).

Can you also clarify the difference between how the marginal likelihood in (3), evaluated using a conjugacy using Poisson and inverse gamma distributions, relates to the nonconjugacy in the "update $z_i$" section of the appendix. Some reference to equation (3) from the main paper in this section of the appendix would also be helpful for clarification.

A few more sentences providing background on the MFM approach might also help clarify this section. What exactly is the form of H?

I think giving the specific model here a name would be useful. This would help the authors more easily reference their specific approach in the paper and later allow future researchers to succinctly refer to the work.

The resolution of the figures is low. Zooming in to see detail is difficult, particularly in 2D.

**Limitations:**

One limitation that I believe is more significant than the author's state is the selection of p being constant across subpopulations. The authors briefly mention ways around this issue in the discussion. I think it is important to more directly point this out as a limitation -- groups of neurons across brain regions very clearly do not have the same dimensionality (low level visual areas and higher level decision making areas, e.g.) This model is only able to isolate subpopulations with this specific constrained feature.

No potential negative social impact of this work.

**Strengths And Weaknesses:**

This paper is technically strong and is an important tool with potential broad use in neuroscience. The addition of a prior over cluster assignment was a non-trivial ML contribution and significant step in understanding increasingly common multi-region datasets with large numbers of neurons. However, there are many pieces to the inference procedure and the appendix has a lot going on. The authors reference many existing approaches and it can be hard to parse the specific approach used in this work.

---

> ### Author Response · Authors · 2022-07-29
> **Response**
>
> Thanks to the reviewer for their positive comments that the “paper is technically strong and is an important tool with potential broad use in neuroscience.” We have taken the reviewer’s suggestion and now refer to our model as “mixDPFA” throughout the text and have fixed the issue with low resolution figures. We have also clarified many specific points…
>
> > Could you expand on how the laplace approximation is being used in 122-124?
>
> The Laplace approximation is being used to generate samples for the MCMC approach, and we have added a sentence around line 122 to make this more explicit.
>
> > An equation showing the P(x|..) ...
>
> The equation for the full conditional distribution P(x|...) is in appendix A.1 (line 487).
>
> > Also it appears there is a typo in the third line of the laplace approximation...
>
> Typo in the third equation for the Laplace approximation in the appendix is fixed, thanks.
>
> > Can you also clarify the difference between how the marginal likelihood in (3), ...
>
> We modified the text in “update z_i” section of the appendix to clarify the point about non-conjugacy.
>
> > A few more sentences providing background on the MFM approach might also help clarify this section.
>
> The motivation and background for MFM are discussed in section 2.2 and line 257-259.
>
> >  What exactly is the form of H?
>
> **H** contains all the prior for cluster parameters (j), which will be complicated when we write things out. The details of prior H for all DPFA parameters can be found in each subsection of A.1 MCMC updates.
>
> > One limitation that I believe is more significant than the author's state is the selection of p being constant across subpopulations.
>
> We also, now expand the discussion about “the selection of p being constant across subpopulations. ” It’s true that “groups of neurons across brain regions very clearly do not have the same dimensionality”. The assumption could certainly limit the accuracy of the model, and we make it more clear that this is an important future direction for the mixDPFA approach.

---

### Meta-Review · Area_Chair_2Pop · 2022-08-30

**Recommendation:** Accept
**Confidence:** Certain

**Metareview:**

The authors present a mixture of dynamic Poisson factor analyzers (sometimes called Poisson linear dynamical systems) model. The model itself is not especially novel (it seems closely related to Poisson switching linear dynamical systems) but the authors make up for it with a well-developed approach to Bayesian inference. They allow for unknown numbers of states with a mixture of finite mixtures model, and they handle the nonconjugacy of the Poisson-Gaussian model with a Metropolis-corrected Polya-gamma augmentation scheme. The empirical results do not show a clear improvement on Neuropixels recordings from the Allen Brain Observatory, but again the assessment is thorough. Overall, I think the paper conveys valuable findings and ideas, and lays a nice foundation for future work. I encourage the authors to incorporate and address the reviewers' feedback when preparing the final manuscript, and to consider expanding the discussion of how the Mix-DPFA model relates to Poisson SLDS.

**Award:**

No

---

### Decision · Program_Chairs · 2022-09-14

Accept